Spatiotemporal Variability and Environmental Controls on Aquatic Methane Emissions in an Arctic
Permafrost Catchment
Michael W. Thayne[1], Karl Kemper[1,2], Christian Wille[1], Aram Kalhori[1], & Torsten Sachs[1,3]
*[1] GFZ Helmholtz Centre for Geosciences, Potsdam, Germany*
*[2] Department of Geosciences, University of Cologne, Cologne, Germany*
*[3] Institute of Geoecology, Technical University of Braunschweig, Braunschweig, Germany*
*Correspondence email: Michael W. Thayne (m_thayne@me.com)*

# Abstract

Understanding spatiotemporal dynamics and drivers of methane ($CH_4$) fluxes from rapidly changing
permafrost regions is critical for improving our understanding of such changes. Between May and
August 2023 and 2024, we measured $CH_4$ using floating chambers in a small Arctic permafrost
catchment on Disko Island, Greenland. Fluxes were derived from 707 chamber measurements using
a semi-automated algorithm incorporating boosted regression trees and generalized additive
models. Highest fluxes occurred in streams and along lakeshores associated with inlets. Diffusive
fluxes dominated (~98% of observations), while only ~1% of chamber deployments exhibited non-
linear concentration increases indicative of ebullition, while the other ~1% were attributed to
uptake. Median diffusive fluxes were 5.0 nmol $m^{-2}s^{-1}$, (-0.1 to 271.8), peaking at ice-break. Ebullition
had a median of 939 nmol $m^{-2}s^{-1}$ (5.2 - 14,893), but did not impact overall fluxes. Model results
suggest that thaw-season $CH_4$ fluxes were initially driven by meteorological conditions and
catchment soil conditions, but shifted rapidly—within approximately one week after ice-off—to
biogeochemical controls, including dissolved organic matter, oxygen saturation, and pH.

# 1 Introduction

Permafrost regions across the Arctic store substantial amounts of carbon. Climate warming
is rapidly changing permafrost regions and consequently their carbon storage dynamics, creating a
critical climate feedback mechanism (Schuur et al., 2015; Miner et al., 2022). At current warming
rates, models project approximately 77% of shallow permafrost will be lost by 2100 (Fox-Kemper et
al. 2021), suggesting large implications for the global carbon budget and how carbon emissions are
distributed across permafrost landscapes. The underlying issue is that thawing permafrost can
release previously frozen organic matter, delivering labile nutrients to soil microbes which accelerate
the decomposition of soil organic carbon as a result of their metabolic processes (Schuur et al., 2015;
Keskitalo et al., 2021; Olefeldt et al., 2021).  Subtle changes in microbial processes in soils can
enhance positive feedback mechanisms which compounds atmospheric warming. Lateral movement
of water through active layer soils is a critical pathway for $CH_4$ emissions from surface waters (Street
et al., 2016; Olid et al., 2021, 2022; Fazi et al., 2021). Hydrological and catchment system dynamics in
particular play a critical role in distributing dissolved carbon throughout permafrost environments.
Catchment systems, such as thermokarst lakes and wetlands, have been shown to be "hotspots" for
$CH_4$ release, where daily emission rates between 10 and 200 mg $m^{-2}d^{-1}$ have been reported (Walter
Anthony et al., 2018; Elder et al., 2020). However, while localized high emissions have been
reported, the overall contribution of Arctic and permafrost freshwater bodies to global methane
budgets is fairly low at 2-6% when compared to other ecoregions such as the tropics at 64%
(Bastviken et al., 2004; Saunois et al., 2025; Virkkala et al., 2024). Nonetheless, with such drastic
change expected, well designed field studies exploring which processes are the most important for
governing CH$_4$ emissions from permafrost regions, are critical data sources for validating climate
models and simulations (Bartsch et al. 2025).
Carbon dynamics in permafrost regions have been shown to be governed by interactions
between soil, vegetation, hydrology, and atmospheric processes (Walter Anthony et al., 2012;
Virkkala et al., 2024; Yuan et al., 2024; Kleber et al. 2025). Located on the central-west coast of
Greenland, Qeqertarsuaq, also known as Disko Island, has become an important data point for
understanding environmental interactions which govern Arctic tundra carbon dynamics. The island
provides a natural laboratory for observing interactions between permafrost, vegetation, microbial
activity and aquatic ecosystems (Humlum, 1998; Humlum et al., 1999; Callaghan et al., 2011;
Christiansen et al., 2015; D'Imperio et al., 2017).  Previous work from the study area has suggested
shifting hydrology, historic permafrost thaw, nutrient cycling, and microbial activity in the active and
permafrost layers as possible drivers of CH$_4$ fluxes from surface water bodies (Zastruzny et al., 2017;
Kluge et al., 2021; Stevenson et al., 2021; Juncher Jørgensen et al., 2024). These studies highlight the
interconnectedness of terrestrial and aquatic ecosystems, and the effect they may have on CH$_4$
fluxes from lakes and streams on Disko Island. There is yet to be an extensive study on CH$_4$ fluxes
from the island's lakes and streams. However, it has been suggested that permafrost thaw and
warming air temperatures may have an effect on greenhouse gas fluxes (Kluge et al., 2021; Juncher
Jørgensen et al., 2024). Soil warming experiments and studies of increased snow cover in winter
were shown to regulate carbon fluxes through accelerated carbon turnover (Ravn et al., 2020; Xu et
al., 2021). Carbon fluxes are further controlled by plant uptake and through microbial activity
regulating the availability of nutrients and subsequent CH$_4$ production (Laanbroek, 2010; Liebner et
al., 2011; D'Imperio et al., 2017). Sedimentary processes in lakes promote carbon storage, whereas
methanotrophic and methanogenic microbial assemblages along an upland–wetland environmental
gradient regulate CH$_4$ consumption and emission, respectively. Therefore, freshwater ecosystems
play a critical role storing, producing, and emitting CH$_4$ (Christiansen et al., 2015; Žárský et al., 2018;
Stevenson et al., 2021).
The hydrology of Disko Island is strongly influenced by past volcanic activity during the
Paleocene epoch. With extensive basaltic lava flows characterizing the landscape, the islands terrain
is formed by the Maligât and Vaigat Formations, which are comprised of highly permeable layers of
basalt interbedded with fluvial and lacustrine sediments (Westergaard-Nielsen et al., 2020; Larsen &
Larsen, 2022). The high permeability of these geologic formations enables substantial subsurface
flow, subsequently forming perennial water features such as warm springs. In spring, the soils which
make up the active layer allow for rapid infiltration of meltwater, which laterally distributes
nutrients and organic matter throughout the island's aquatic ecosystems (Westergaard-Nielsen et
al., 2020). For example, during spring runoff meltwater and hillside topography was found to largely
drive the distribution of nitrates from terrestrial to aquatic ecosystems (Zastruzny et al., 2017;
Stevenson et al., 2021). Thus, pools of nutrients available during the growing season may vary
dramatically from one year to the next. Lateral flow of snowmelt and permafrost thaw may influence
CH$_4$ fluxes due to changes in physio- and biogeochemical properties of the lakes, streams and rivers
on the island (Liebner et al., 2011; Rautio et al., 2011; Walvoord & Kurylyk, 2016; Stevenson et al.,
2021). Although Disko Island has discontinuous permafrost (Christiansen et al., 2015; Kluge et al.,
2021), thawing can release trapped organic matter and nutrients into aquatic ecosystems,
potentially affecting CH$_4$ fluxes by providing new substrates for microbial activity (Ravn et al., 2020;
Stevenson et al., 2021; Westergaard-Nielsen et al., 2020; Xu et al., 2021).
The distribution and drivers of aquatic CH$_4$ emissions in permafrost regions remain poorly
constrained, particularly across small lakes and streams which may arise as emission hotspots.
Previous studies on Disko Island have highlighted the potential importance of hydrology, permafrost
thaw, and microbial processes for greenhouse gas fluxes, but comprehensive spatial and seasonal
assessments of $CH_4$ are lacking. In this study we address this gap by quantifying $CH_4$ fluxes from 707
floating chamber measurements across a permafrost-affected catchment (Sanningasup Tasia). Using
boosted regression trees, we evaluate the partial effects of physiochemical water conditions,
catchment soil conditions, and meteorology in regulating emissions from ice-break through the
growing season. Our objective was to determine how spatial heterogeneity and seasonal dynamics
shape $CH_4$ emissions from Arctic freshwater ecosystems and to identify the key processes that
control flux variability in permafrost catchments.

# 2 Methods

## 2.1 Study Site

Lake Sanningasup Tasia in Greenlandic, or Moræne sø in Danish, is situated between
moraines in the north and east and an outlet which drains into the Red River to the west (Figure 1).
The lake is primarily fed by a large warm spring which enters the lake from the southeast, forming a
wetland type ecosystem. The other inlets of the lake are primarily fed by seasonal snowmelt. The
heterogeneity of the catchment provides an exemplary study site, allowing us to understand the
mechanisms regulating $CH_4$ emissions from a lake, streams, and wetland.  According to a 2018 report
from the University of Copenhagen, the lake has a maximum depth of 4.5 m and is generally
phosphorus limited with nitrogen concentrations being seasonally variable, where concentrations
during ice cover are higher than during periods of no ice cover (Westergaard-Nielsen et al., 2020).
We found water temperature of the lake to range between 1.1 and 13.9 $^o$C with a mean of 7.9 $^o$C. To
our knowledge there has never been an extensive study on the greenhouse gas fluxes from the lake
and surrounding water bodies.


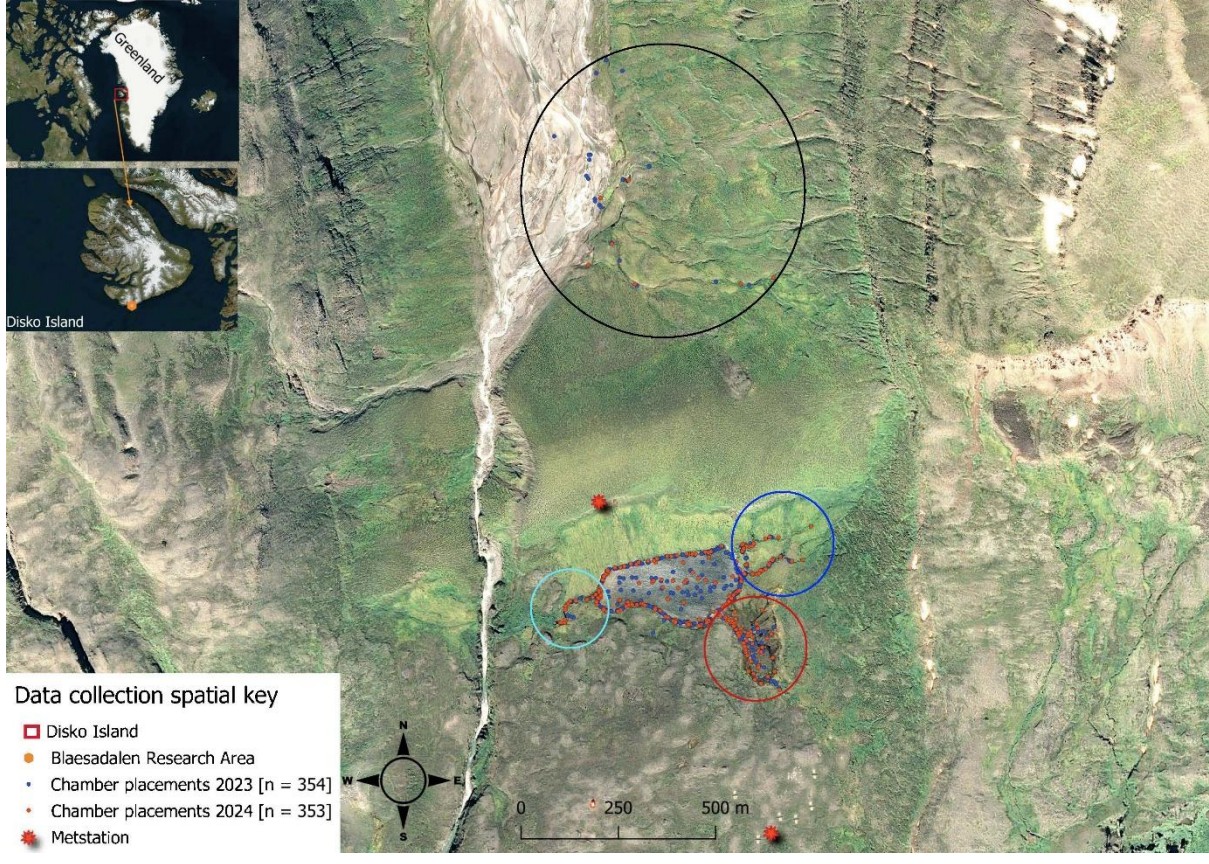

Figure 1. Map showing the 707 chamber measurements (blue and brown dots). Points in the south
are concentrated around Lake Sanningasup Tasia and its connected streams. The blue circle indicates
streams fed by snowmelt and the turquoise circle indicates the outlet of the lake. The red circle
indicates the warm spring area which forms a wetland type ecosystem. Points inside the black circle
north of the lake indicate measurements taken from the Red River and its stream tributaries.
Orthomosaic background image © CNES (2024), Distribution Airbus DS, produced from Pléiades 1B
satellite imagery.

## 2.2 Data Collection

We used closed floating chamber systems connected to an ABB/Los Gatos Research GLA131
Series Micro portable Gas Analyzer in 2023, and to a LI-COR® LI-7810 Trace Gas Analyzer in 2024,
with the goal of capturing the spatial and temporal variability of $CH_4$ fluxes in the catchment area. In
2023, we used a self-built cylindrical chamber made of semi-transparent plastic material with
volumetric capacity of 0.016 $m^3$ and a basal area of 0.096 $m^2$. In 2024, we used a West Systems type
C flux chamber made of stainless steel with a volumetric capacity of 0.013 $m^3$ and a basal area of
0.07 $m^2$. Both chambers included a fan for mixing air and a temperature sensor. The semi-
transparent plastic chamber used a circular foam floater that was wrapped around the outside of
the chamber, allowing 2cm of the chamber to be below the water surface, forming a 100% airtight
seal. The West Systems chamber was inserted into a closed-cell foam floater, where the seal was
created once the chamber was inserted into the floater. Despite differing chamber materials and
flotation devices, median fluxes between 2023 and 2024 were identical at 5.0 nmol $m^{-2}s^{-1}$. However,
to evaluate whether changes in chamber construction between years introduced systematic bias in
$CH_4$ flux estimates, we fit a linear model using log-transformed flux as the response and chamber
type, latitude, longitude, and Julian day as predictors. Chamber type was not a significant predictor

($p$ = 0.13), and QQ plots of log-transformed fluxes across years showed no consistent deviation
across the flux distribution, except at lower emission rates (Figure S1). These findings suggest that
differences in chamber construction did not substantively influence the calculated fluxes. Chamber
measurements were conducted on the surface of the lake and surrounding water bodies at a spatial
distance of 10 to 20 m with a closure time of 10 minutes. The chamber and gas analyzer were
connected in a closed loop, and sample air was continuously pumped through the gas analyzer. $CH_4$
concentrations were measured with a frequency of 1 Hz. Each flux measurement was given a unique
id based on its spatial location or water body type. For the lake, ids were given based on what
shoreline we were measuring on. For example, if on the east shore, ids would be e1, e2, e3, … etc.
The majority of flux measurements on the lake were conducted within 0.2 m from the shore. Open
water measurements in the lake were made using a small boat and anchor system. $CH_4$
concentrations in streams were measured by starting at, or near the headwaters and then taking
measurements progressively downstream with consideration to the changing terrestrial vegetation
and stream dynamics (i.e., fast, slow, or eddie pool). In 2023, we measured isolated meltwater pools
during the thaw to represent control conditions (water not yet connected to the lake or streams). In
2024, to capture an even earlier baseline, we conducted chamber measurements on top of snow
and lake ice prior to thaw onset, providing a true pre-thaw control period. Overall, we made 707
chamber measurements, representing ~10-15 chamber placements taking place daily, around the
lake and surrounding streams giving us an extensive view of the spatial and temporal variability of
$CH_4$ fluxes in the study area. While floating chambers isolate the headspace from light wind
disturbance, increased surface turbulence may influence gas exchange in the open space of the
chamber (Vachon and Prairie 2013). Our approach captures diffusive exchange under mostly
calm-water conditions (i.e., wind speed up to ~4 ms$^{-1}$), but we acknowledge that regional wind-
driven mixing may contribute to flux variability beyond individual chamber footprints. We
simultaneously measured water temperature using Truebner EC-100 RS-485 EC/Temperature
sensors in 2023 and a suite of water parameters were collected in 2024 using an AquaTroll 600
water sonde (see section: Decoding Methane Drivers). Meteorological data and soil characteristics
were collected from nearby meteorological stations maintained by Aarhus University which are part
of the Greenland Ecosystem Monitoring Database (Greenland Ecosystem Monitoring, 2025a-d) (see
section "Decoding Methane Drivers" for list of variables used).

## 2.3 Flux Algorithm & Ebullition Detection

In collaboration with ChatGPT 4.0, we wrote an interactive algorithm in R which leverages
General Additive Models (GAM) and Boosted Regression Trees (BRT) to robustly and flexibly
calculate $CH_4$ fluxes from individual floating chamber measurements (Figure 6). The flux calculation
procedure was applied identically to individual chamber time series for both 2023 and 2024, while
controlling for different the chamber constructions.

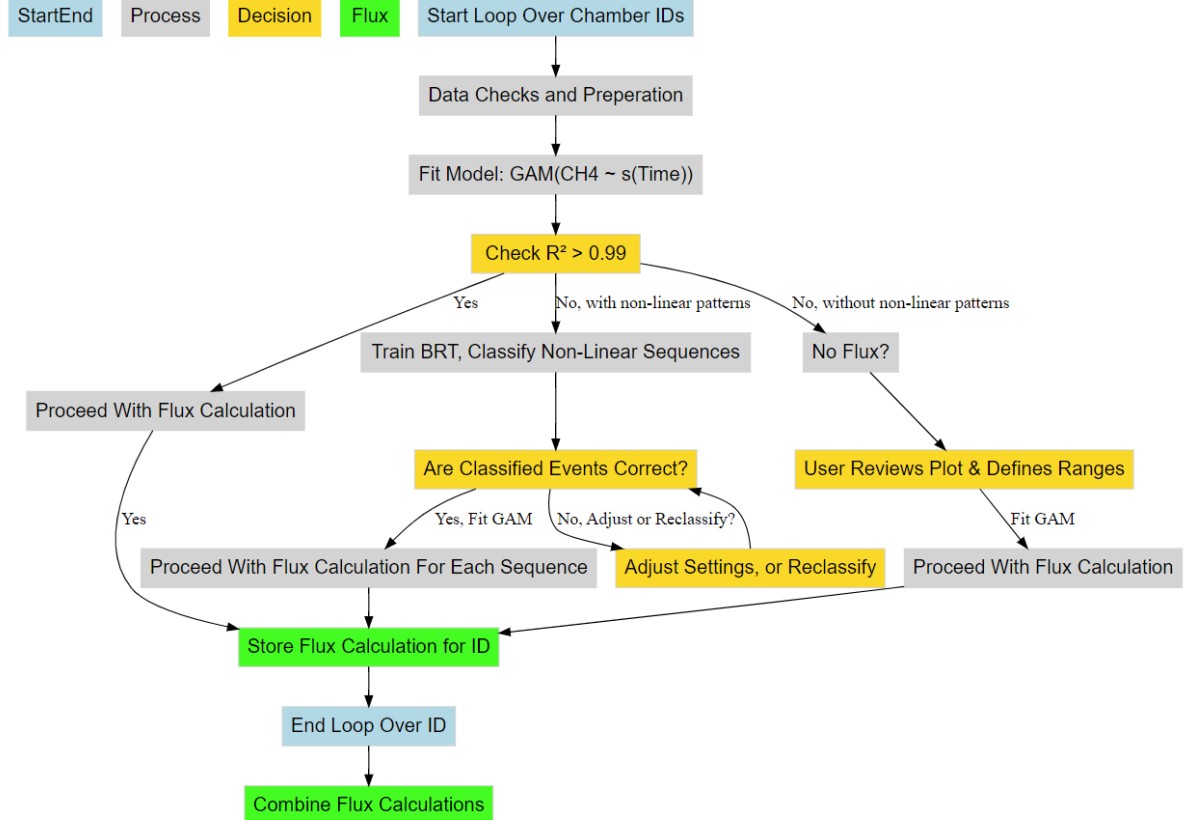


Figure 2. CH$_4$ flux calculation workflow from concentration data using predictions from GAM and BRT. Blue boxes represent the start and end of a single chamber measurement working through the algorithms processes (grey boxes) and decision logic (gold boxes). Green boxes represent the storage and combination of the results for further analysis.

Rather than fitting concentration data with linear, exponential, and/or polynomial models (Kutzbach et al., 2007; Pedersen et al., 2010; Hoffmann et al., 2017), the algorithm fits GAM, which are capable of modelling non-linear patterns without a-priori specification of the functional form of the relationship between time and concentration. However, before fitting a GAM, the concentration and accompanying data is checked and processed (Figure 2; "Data Checks and Preparation") as follows: The algorithm conducts a preliminary check for the required chamber parameters which are; id, ordered times of measurement, air temperature (°C), volume, area, and air pressure. In addition, air temperature is expected to be initially in Celsius, which is automatically converted to Kelvin during the processing of concentration data in preceding steps. The data is then ordered based on id and time to maintain correct chronological order of chamber measurements. CH$_4$ is then converted from ppm to moles using Ideal Gas Law:


$$CH4_{moles} = \frac{(CH4ppm \times P \times V)}{(R \times T)} \qquad (1)$$


where P is air pressure (Pa), V is chamber volume (m$^3$), R (8.314 J/ (mol × K) is the universal gas constant, and T is air temperature inside the chamber (K). After preliminary data checks and initial

processing of the concentration data, the concentration time series is then fit to a GAM (i.e.
gam(CH4$_{moles}$ ~ s(time, k = gam_knots)), where the smoothing parameter 'k' is a user defined
parameter named 'gam_knots' which has a default value of 5. The value of 'k' cannot exceed 3 times
the degrees of freedom for a given concentration time series, or the algorithm defaults to fitting a
linear model. The algorithm then checks the $R^2$ value of the fitted GAM to see if it meets the default
conditional value of ≥ 0.99, if so, it proceeds to calculate fluxes following these steps:

$$\Delta CH4_{moles} = \left[ \frac{CH_4(t_2) - CH_4(t_1)}{t_2 - t_1}, \frac{CH_4(t_3) - CH_4(t_2)}{t_3 - t_2}, \dots \frac{CH_4(t_n) - CH_4(t_{n-1})}{t_n - t_{n-1}} \right] \quad (2)$$

Where $\Delta CH4_{moles}$ is the rate of change, calculated as the quotient of the predicted differences in
$CH_4$ concentration between successive time points. Flux is then calculated between successive time
points by:

$$\bar{F}_{Flux} = \frac{1}{n-1} \sum_{i=1}^{n-1} \frac{\Delta CH_4}{A} \quad (3)$$

Where mean flux of the chamber measurement is estimated by calculating flux at each
successive time step, where flux is determined by dividing $\Delta CH4_{moles}$ by the basal area (A) of the
chamber, expressed in ($m^2$).  A plot of the time series and model fit is generated and saved in the file
directory defined by the user by setting the parameter 'save_directory' (Figures S1-5). Because
fluxes were derived from high-frequency (1 Hz) concentration data fitted using a GAM with a strict
acceptance threshold of $R^2 \geq 0.99$, analytical uncertainty in the rate-of-change estimation is
negligible. Conventional uncertainty propagation (e.g., based on regression slope error or replicate
chambers) is not meaningful in this context because the GAM approach fits a smooth curve through
hundreds of data points per deployment, effectively minimizing noise and preventing poor-quality
fits from contributing to the final flux values. This ensures that the dominant source of variability in
the dataset reflects true environmental heterogeneity rather than analytical error. Furthermore,
because the chamber headspace was fully sealed and isolated from external turbulence, wind-
induced variability—which often motivates uncertainty corrections—is mechanically removed from
the flux calculation process. For these reasons, we report spatial variability (e.g., medians, ranges,
and interquartile spread) rather than analytical uncertainty, as it provides a more ecologically
relevant representation of flux variability across the catchment.
In the cases where the initial GAM fit does not meet the $R^2 \geq 0.99$ condition, the algorithm
can follow two pathways (Figure 2). Pathway (1) is a result of the algorithm having detected non-
linear concentration increases using BRT, while pathway (2) the algorithm has found the chamber
measurement has not met any of the conditional requirements for flux calculations, or more
generally stated, there was no measurable concentration increase detected automatically. Both
pathways are interactive as the user is prompted to confirm the classification of "ebullition" versus.
diffusive data sequences in pathway (1), while in pathway (2) the user confirms there is indeed no
concentration increase by reviewing the diagnostic plots (see Supplemental text and Figures S2-S6).
Once confirmed, the user initiates flux calculations by manually entering the time range of the
measurement that should be fitted (Figure 2). "Ebullition" in the context of the algorithm refers to a
sudden, non-linear CH$_4$ increases identified by the algorithm, which likely includes ebullitive events,
but does not strictly infer all fluxes calculated this way were from bubbles entering the chamber.

## 2.4 Lake and Stream Metabolism

In 2024, we calculated the metabolic parameters net ecosystem production (NEP), gross
primary production (GPP), and ecosystem respiration (ER) of oxygen for the lake and streams using
dissolved oxygen (DO) collected at a one-minute frequency during chamber measurements (DO
sensor accuracy: ± 0.1 mg L$^{-1}$; resolution 0.01 mg L$^{-1}$). Including metabolic parameters as predictors
in the BRT models gave us an understanding of the role microbial oxygen production plays in
regulating or not CH$_4$ emissions from water. DO saturation was adjusted using temperature-
dependent solubility constants (Garcia & Gordon, 1992). Oxygen flux at the air-water interface was
determined using wind derived gas exchange coefficients and adjusted for water temperature (Cole
& Caraco, 1998). NEP was calculated as the rate of change in DO concentration over each chamber
deployment, adjusted for air-water exchange (Hall & Madinger, 2018; Noss et al., 2018). GPP and ER
were partitioned from NEP by applying a threshold of 200 µmol m$^{-2}$ s$^{-1}$ photosynthetically active
radiation (PAR), which distinguishes intervals with effective photosynthesis from those with
negligible light-driven production, despite continuous daylight during Arctic summer. Aggregating
each chamber DO measurement into one-minute intervals, we calculated mean NEP, GPP, and ER for
each chamber placement (Winslow et al., 2016). While this methodology is sound for the lake, there
is some caveats in relation to stream metabolism. Because we are using a model which assumes
wind driven gas exchange for small streams, we likely underestimate gas exchange in parts of the
streams where turbulence from streambed roughness dominates. Nonetheless, the approach
captures broadly the metabolic trends in lake and stream metabolism observed in other Arctic and
Boreal waterbodies (Mulholland et al., 2001; Rocher-Ros et al., 2021; Ayala-Borda et al., 2024; Klaus
et al., 2022; Myrstener et al., 2021) and is useful for comparing fluxes across aquatic biomes.

## 2.5 Spatial Flux Evaluation

We uploaded as a spatial layer in QGIS version 3.40.1 (QGIS Development Team, 2025) an
orthomosaic image produced by Airbus satellite Pléiades 1B and the chamber placements as points
with their associated flux estimates. We spatially analyzed the CH$_4$ fluxes by creating bi-weekly
emission heatmaps using the Kernal Density Estimation (KDE) algorithm in QGIS. The use of KDE
allowed us to smooth across discrete chamber measurements, yielding an intuitive continuous
surface representation of CH$_4$ flux hotspots and their evolution through time (Figure 2). We set the
radius to between 30-35 meters to allow some connectivity between points which allows for a
smooth representation of any environmental gradients that might be captured. We used the default
method using a quartic kernel shape weighted by the flux calculated for each chamber placement.
While "ebullitive" fluxes were not considered in further statistical analysis (i.e., in the BRT), those
fluxes are depicted in the resulting heatmaps. Overall, using KDE allowed for an intuitive
interpretation of the seasonal and spatial development of flux hotspots in the research area.

We additionally compared daily CH$_4$ emissions from Sanningasup Tasia catchment relative to
other Arctic-Boreal Lake classes compiled in the Boreal-Arctic Wetland and Lake Dataset (Kuhn et al.
2021; Olefeldt et al., 2021). A Kruskal-Wallis test was performed to determine significant differences
($p < 0.05$) in the log transformed daily fluxes from Sanningasup Tasia (n = 48) and its streams (n = 35)
relative to broader biome-scale fluxes from Small Peat Lakes (n = 50), Medium Peat Lakes (n = 36),
Large Lakes (n = 10), and Small Yedoma Lakes (n = 7). We then performed pairwise Wilcoxon rank-
sum tests with Benjamini-Hochberg adjustment for multiple comparisons to highlight specific
contrasts between Sanningasup Tasia catchment and the other lake classes. Conducting this
provides an understanding of how Sanningasup Tasia catchment emissions compares to other Arctic
waterbody types.

## 2.6 Decoding Methane Drivers

To determine important drivers and their partial effects on diffusive $CH_4$ fluxes, we trained
BRT with physicochemical water parameters, catchment soil temperatures, catchment soil moisture,
surface air temperatures, local meteorology, and Julian day (Figure 3). Considering we only collected
water temperature in 2023, we used flux data from 2024 for training the BRT. We have focused on
diffusive fluxes due to the unpredictability of fluxes when "ebullitive" processes were considered in
the models. The diffusive fluxes give us a detailed view of environmental controls shaping $CH_4$ fluxes
coming from the catchment. To characterize fluxes we assembled a comprehensive set of predictors
including; (a) aquatic variables measured *in-situ* with a water sonde at each chamber deployment
(e.g., Conductivity (µS/cm), pH, redox potential (mV), dissolved oxygen (mg/L), oxygen saturation
(%), water temperature (°C), and fluorescent dissolved organic matter (FDOM; RFU)) (Figure S7), (b)
catchment soil characteristics collected at nearby climate stations (e.g., soil volumetric water
content at 10 cm and soil temperature at 40 cm), and (c) meteorological variables collected from a
nearby climate station (e.g., Surface air temperature at 2 cm (°C), Air temperature (°C), relative
humidity (%), air pressure (mbar), precipitation (mm), PAR ($\mu mol\ m^{-2}s^{-1}$), and mean wind speed ($ms^{-1}$)
and direction (°)). Lake water levels (mm) were included to characterize the effect of changing
hydrologic conditions and its influence on lake $CH_4$ fluxes. Although water depth was recorded
during chamber deployments using the AquaTroll 600, it was not included as a predictor in the BRT
models. In shallow Arctic lakes like Sanningasup Tasia (<4.5 m), $CH_4$ is primarily sediment-derived,
with deeper zones more likely to promote oxidation or dilution due to greater oxygen exposure
(Bogard et al., 2014; Bulínová et al., 2025; Emerson et al., 2021; Li et al., 2020). Shoreline fluxes
often dominate due to anoxic, vegetated sediments, while interior zones tend to suppress emissions
(Thompson et al., 2016; Kyzivat et al., 2022; Rasilo et al., 2015). We therefore prioritized
biogeochemical water column predictors—FDOM, dissolved oxygen saturation, and GPP—over
depth (Christiansen et al., 2015; Singleton et al., 2018), and explicitly captured depth gradients via
boat-based chamber deployments across the lake interior. Catchment soil characteristics were
included to capture the hydrogeological conditions surrounding the catchment. We used catchment
soil temperature at 40 cm to represent subsurface active-layer conditions that influence deeper
thermal dynamics, groundwater inflow, and delayed soil heat retention through the thaw
season. Soil volumetric water content (VWC) at 10 cm was included to gain an understanding if
dryer, or wetter catchment conditions effect surface water $CH_4$ fluxes, and to act as a substitute for
water level in the lake early in the season as these two share a Pearson's correlation of r = 0.93.
Additionally, we used VWC at 10 cm depth because it was the most complete and continuous
dataset across the measurement depths, and highly correlated with VWC at 20 cm, 30 cm and 40
cm. To reduce multicollinearity amongst the predictors, we set up weighting for random feature
selection by calculating the average absolute Pearson correlations between predictors and assigning
weights inversely proportional to the correlations, resulting in higher weights given to predictors
with decreased collinearity and thus more likely to be included as a predictor.

Using the "gbm.step" algorithm from the R package "dismo" version 1.3.14 (Elith et al.,
2008; Hijmans et al. 2023), we iteratively attempted to fit 500 BRT with a subset of randomly
sampled two-week time series of flux calculations and 7 of the 21 weighted predictors. Each two
weeks must have at least 90 observations, or the date range is buffered on either end of the time
series to meet the minimum observation requirement. The algorithm uses 10-fold cross-validation to
minimize overfitting the models (Elith et al., 2008). If a randomly sampled two weeks did not meet
the minimum required observation of 90, the time series was buffered on both ends of the date
range to meet the minimum required observations. BRT were optimized using a grid search where
hyperparameters such as learning rate (0.001, 0.002, 0.003, 0.004, 0.005), tree complexity (1, 3, 5,
7), and bag fraction (0.30, 0.40, 0.50), were tuned for each model fit. While bag fraction values in the
range of 0.5–0.8 are more commonly used, a lower bag fraction increases stochasticity in tree
construction, which helps reduce overfitting—especially important for modeling noisy and highly
non-linear $CH_4$ flux data. This conservative approach favors identifying robust general patterns
rather than fitting noise or outliers. Variable monotonicity was handled dynamically for each subset
of predictors using Spearman's rank correlation. Monotonicity for categorical variables was set to
zero, while positive correlations were given a +1 and negative correlations were given a -1. The
model with the best composite score was selected for each iteration. The composite score was
calculated by adding together the standardized cross-validation error, standardized correlation
error, and the cross-validation correlation. The model was finally saved after checking for over fitting
by taking the difference between cross validated mean deviance and training mean deviance and
dividing the difference by training mean deviance. Because the inherently noisy nature of ecological
data, we allowed for 40% difference between cross validated predictions and training data. We
further calculated the percent deviance explained for each BRT model using the formula: % deviance
explained = 100 × ((null deviance − residual deviance) / null deviance), where the null deviance
represents the deviance of a model using only the mean response, and the residual deviance is from
the fitted BRT model. Each fitted model and its metadata were saved for further analysis. This
modelling structure ensures robustness against outliers and ensures data integrity through
dynamically handling monotonicity and applying overfitting constraints. Furthermore, the structure
ensures robust predictions of fluxes by accounting for multi-collinearity amongst predictors and flux
heterogeneity throughout the season.

To visualize the results of the models, we plotted partial dependence two ways. First, we
extracted partial dependence information for understanding model structure, i.e., those predictors
and interactions which were used to split trees and decrease cross validated prediction error. In
addition, we made isolated predictions for each environmental feature in the model by holding all
other predictors at their median to gain a more mechanistic understanding of which conditions
and/or processes are directly affecting $CH_4$ fluxes. The two ways of visualizing partial dependence
give us an ecological understanding of how integrated direct and indirect effects regulate fluxes from
the catchment, but also how individual variables and/or processes regulate fluxes from the water
surfaces, respectively. All visualizations were generated using the R package "ggplot2" version 3.5.0
(Wickham, 2016), and the package "DiagrammeR" version 1.0.11 for flowcharts (Iannone, 2024).

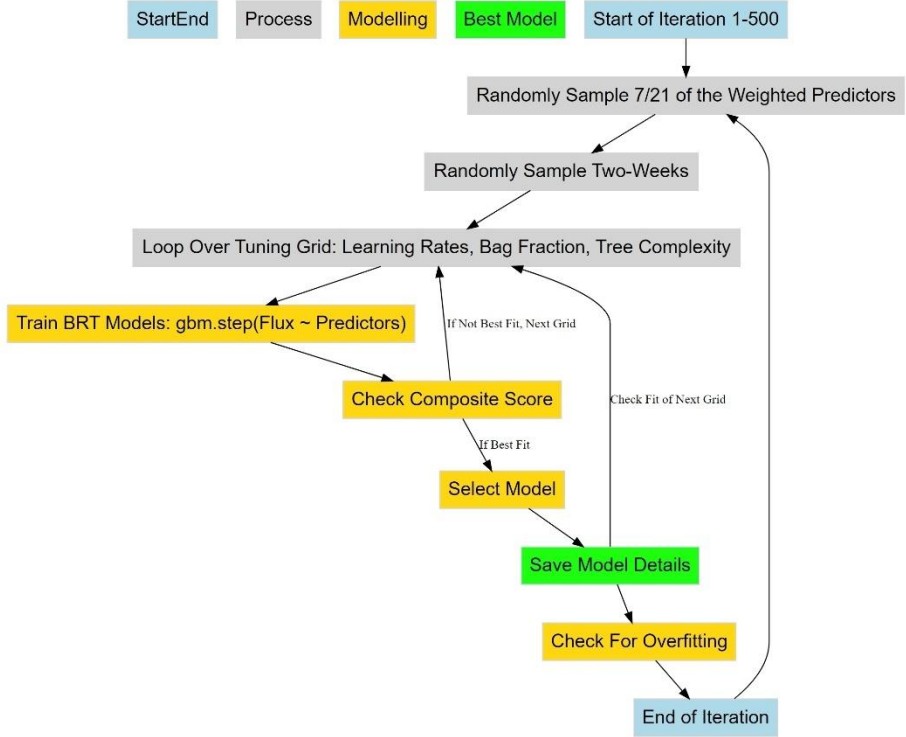


Figure 3. Workflow of the iterative randomized process for selecting the best BRT for predicting diffusive $CH_4$ fluxes using various environmental predictors. Blue boxes represent the start and end of a single iteration through an index of weighted predictors, time periods, and tuning grid (grey boxes). Yellow boxes represent the model selection logic, while the green box represents the storage of flux predictions and selected model details for the included predictors and two-week sub-sample of chamber measurements.

# 3 Results and Discussion

## 3.1 Overview

Methane fluxes from permafrost affected catchments are influenced by a complex interplay between climatological, hydrogeological, and biogeochemical processes. This study highlights the transient nature of $CH_4$ fluxes from a permafrost catchment in west central Greenland and the partial effects of physiochemical water conditions, local meteorology and catchment conditions (Figures 4-7). $CH_4$ emissions from the catchment were variable across water conditions, with streams exhibiting the highest emissions (Figure 4). In comparison to the global coverage of the Boreal-Arctic Wetland and Lake Dataset (Olefeldt et al., 2021), besides small peat lakes, daily fluxes from Sanningasup Tasia at 8.3 mg m$^{-2}$ d$^{-1}$ were mostly comparable to other permafrost waterbodies across the Arctic-Boreal region, which ranges between 3.8-5.4 mg m$^{-2}$ d$^{-1}$ (Figure 4). Highlighting the importance of emissions from small streams, Sanningasup Tasia streams showed significantly higher daily fluxes (18.2 mg m$^{-2}$ d$^{-1}$) than all inland water body classes, except Yedoma lakes (43.7 mg m$^{-2}$ d$^{-1}$). Our results indicate that $CH_4$ fluxes were seasonally variable and controls on fluxes shifted from hydroclimatic factors during colder periods to biogeochemical processes as the catchment warmed and increased in productivity (Figure 3-4 and Figure A1). The seasonal thaw of annual snow and ice accumulation in the two study years varied in timing and duration due to 2023 staying anomalously snowy until the beginning of July, where in 2024 the number of snow free days aligned with historical records. In 2023, our initial chamber measurements between July 03-15 captured peak

median fluxes at 8.9 nmol m$^{-2}$s$^{-1}$ just as the ice began to break on the lake. In an effort to capture similar conditions in 2024, we used an index of historical snow free days on the island which led to us capturing median fluxes of 0.18 nmol m$^{-2}$s$^{-1}$ between May 24-June 05 atop ice and snow. Peak median fluxes of 8.1 nmol m$^{-2}$s$^{-1}$ in 2024, were comparable to 2023, but occurred a month earlier between June 11-19. As runoff water receded and the catchment warmed, growing season commenced in conjunction with steadily decreasing median fluxes between 3.9 and 4.5 nmol m$^{-2}$s$^{-1}$. This study further provides methods to disentangle important drivers and their partial effects on CH$_4$ fluxes using BRT (Figure 3-4). In general, CH$_4$ fluxes were strongly dependent on discrete oxic-anoxic aquatic environments under the chamber (Figure 4 and Figure A1). This research emphasizes the importance of integrating field measurements with GIS-based spatial analysis to monitor CH$_4$ fluxes in permafrost catchments. It further drives home the importance of seasonal transition periods in predicting fluxes from Arctic waterbodies.

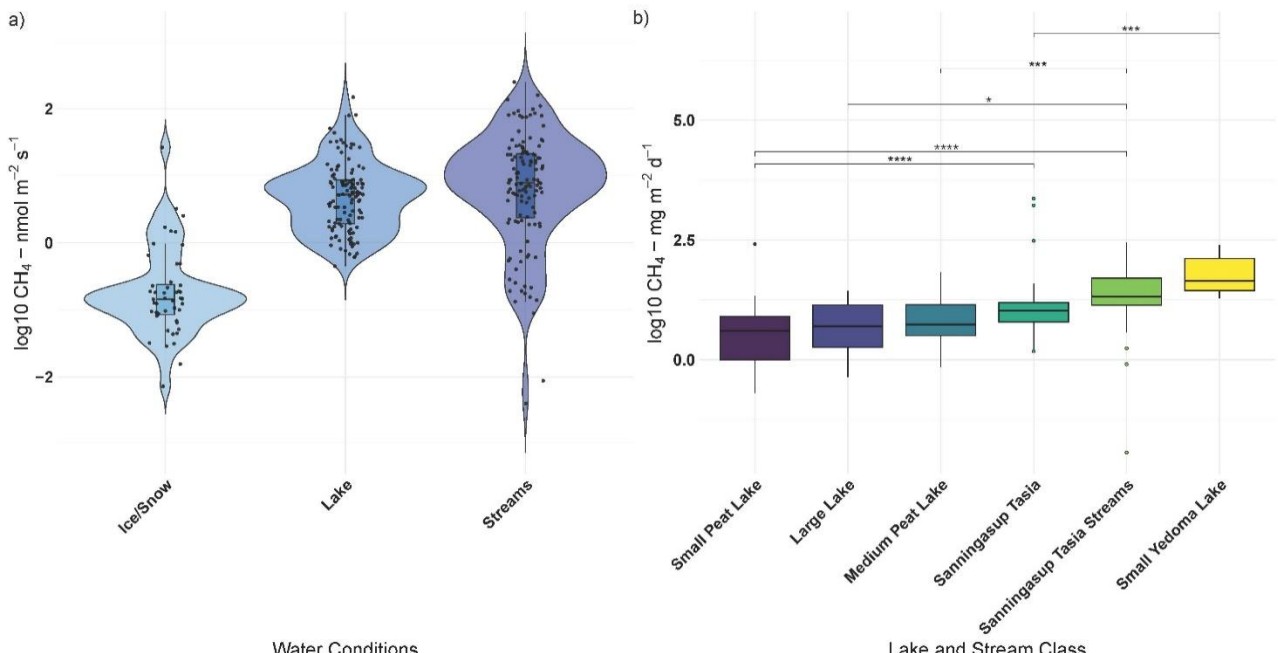

Figure 4. Panel a) shows log transformed CH$_4$ fluxes across the different catchment water conditions during 2024 field season and b) compares log transformed daily CH$_4$ fluxes (y axis) between Sanningasup Tasia catchment and other permafrost waterbodies (Kuhn et al., 2021) across the Arctic-boreal region. Connecting brackets and stars show, for example, that Sanningasup Tasia had significantly (p < 0.05) higher daily emissions when compared to Small Peat Lakes, while daily emissions from Sanningasup Tasia Streams were significantly higher than all lake classes except Yedoma Lakes.

## 3.2 Spatial and Temporal Evolution of Methane Fluxes

In both 2023 and 2024, spatial and temporal evolution of fluxes occurred somewhat heterogeneously in the catchment, but generally "hotspots" occurred in the streams and where they enter the lake (Figure 5 a-b, e-f). Despite different time periods of the thaw, fluxes in the catchment in both years followed a similar trajectory, with peak fluxes occurring post thaw and decreasing through the growing season. In 2023, snow persisted anomalously late into the summer season, and soil temperatures were the coldest recorded in a 6-year record (Figure S8). We found that local climate and catchment soil characteristics were at times, (i.e., during the thaw season and towards the peak of growing season) more important than water temperature in predicting 2023 fluxes, suggesting catchment contributions to surface waters plays an indirect role in CH$_4$ fluxes (Figure S9).

While water temperature was found to be relatively important in both years, catchment CH₄ fluxes in 2024 suggest the system is more driven by variability in dissolved organic matter and microbial production of oxygen (Figure 4-7 and Figure A1).

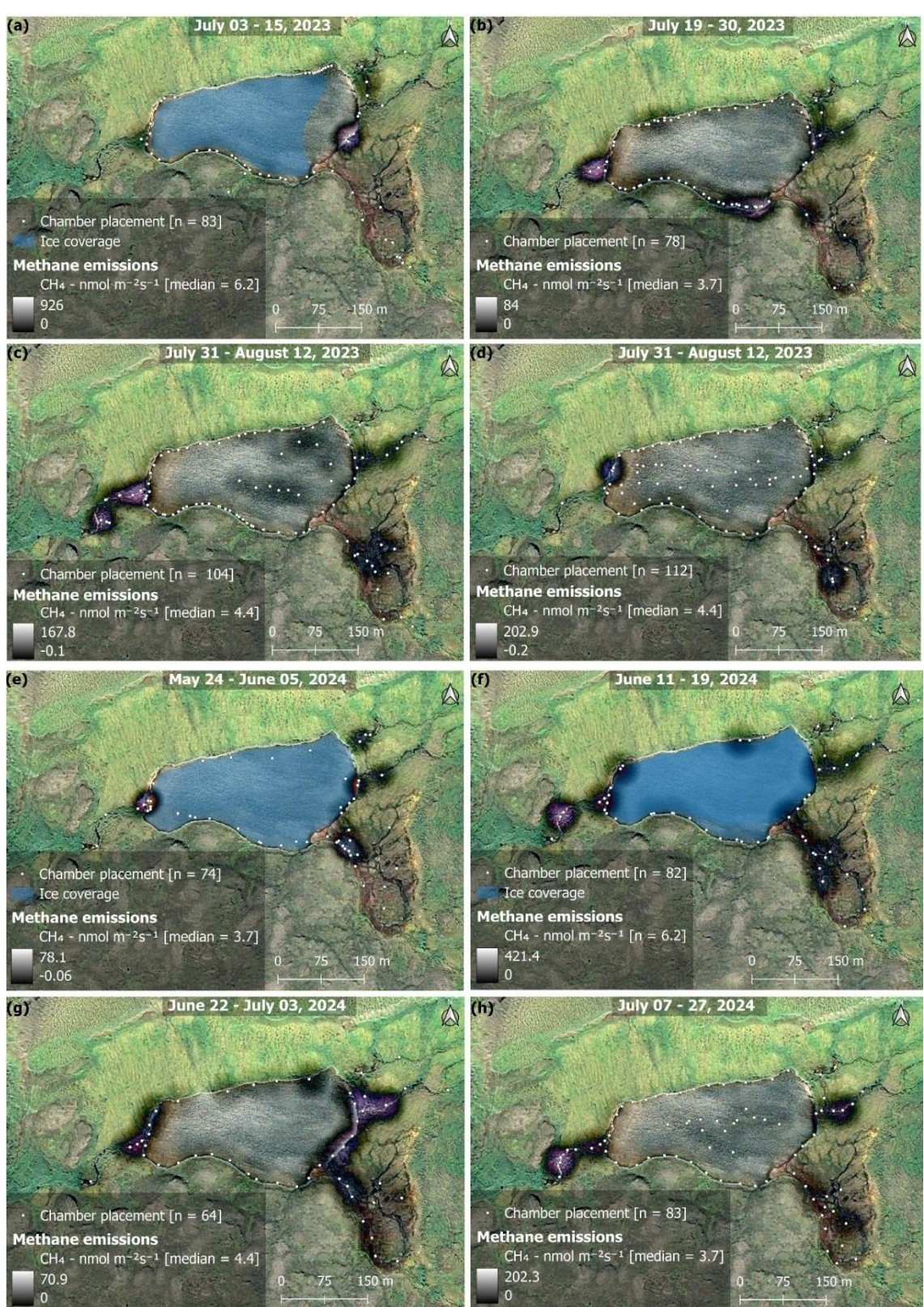

Figure 5. Maps showing the bi-weekly spatial evolution of CH₄ emissions during 2023 (a-d) and 2024 (e-h). The time period covered in each map is given at the top of each map. Whiter colors represent

areas of high emissions, while darker, or no color represent areas of lower, or no emissions (see
color bars in map legends). Areas outside of the water were not measured and are artifacts from the
KDE mapping algorithm. Each white point on the map represents the placement of the floating
chamber. Orthomosaic background image © CNES (2024), Distribution Airbus DS, produced from
Pléiades 1B satellite imagery.
The main inlet to the lake is a warm spring with median temperatures of 7.4 °C and tended
to be a persistent location throughout the season for increased CH$_4$ fluxes. The warm spring area in
the southeast is a complex area where a perennial spring bubbles out of the ground forming a
perennial tributary to the lake. However, the spring seeps out along the base of the hillslope to the
east, subsequently forming a peat fen environment (Figure 5). The eastern most inlets are fed by
meltwater flowing through the vadose zone, but also served as a persistent location for increased
CH$_4$ fluxes. Fluxes from inlet streams followed along an upland-wetland environmental gradient,
where fluxes at the headwaters of streams were generally close to 0, or slightly taking up CH$_4$, but
steadily increased as steeper upland slopes gave way to more gradual water saturated slopes and
pooling sediments. This is consistent with previous work on the island showing CH$_4$ fluxes association
with soil microbial assemblages shifting from methanotrophic to methanogenic along an upland-
wetland gradient, respectively(Christiansen et al., 2015). Additionally, the catchment serves as
micro-topography previously described as an area of snowmelt retention, and subsequently an area
of nutrient and/or dissolved organic matter pooling (Westergaard-Nielsen et al., 2020), which has an
impact on CH$_4$ fluxes throughout the season (Figure 7). While high fluxes were recorded along the
shore and in the open water of the lake, fluxes tended to be patchy and decreased moving away
from the inlet streams (Figure 5 and Figure S10). However, as soon as the lake water flowed to the
outlet, fluxes increased substantially. As the summer season progressed, CH$_4$ fluxes declined across
most of the catchment, becoming largely confined to the warm spring inlet and the eastern inlet
streams (Figure 5). Field observations of late-season fluxes in 2024, found decreased fluxes were
associated with submerged filamentous green algae in stream channels, while assemblages of iron-
oxidizing bacteria on the stream banks were associated with increased fluxes, (Figure A2). The
spatial and temporal evolution of fluxes was driven by seasonally shifting environmental conditions.

### 3.3 Boosted Regression Tree Results

Out of 500 iterations, 321 BRT were fit, and showed good alignment and consistently
performed well in cross-validation, with a correlation median of 0.40 between observed and
predicted values, and a median deviance standard error of 131. Between 8.4% and 62.4% with a
median of 27.3% of the CH$_4$ flux variability was explained by the various models and included
environmental conditions, suggesting a substantial proportion of CH$_4$ fluxes were explained by the
environmental conditions included. The calculated root mean squared error (RMSE), which reflects
the average magnitude in prediction error of the BRT, ranged between 6.5 to 28 nmol m$^{-2}$s$^{-1}$, with a
median of 13.7 nmol m$^{-2}$s$^{-1}$. The summary statistics reflect models that performed reliably and with
fairly good accuracy in predicting diffusive CH$_4$ fluxes from the catchment in 2024. The models
predicted shifting relative importance (Figure 6) and partial effects of the various environmental
conditions throughout the season (Figure 7). The magnitude of CH$_4$ fluxes predicted by the BRT
models were strongly influenced by localized biogeochemical conditions within the water column
based on whether the flux was originating from the lake, stream, or if it was influenced by ice or
snow (Figure 4-7, and Figure A1a-d). Visualizing partial dependence of predictors important for
model structure, revealed integrated ecological effects between local meteorology, catchment
conditions and physiochemical water conditions (Figure 7). However, isolated direct marginal effects
of the various environmental conditions suggest fluxes from water surfaces are directly regulated via
biochemical processes associated with GPP and ER of oxygen (Figure A1b-d).

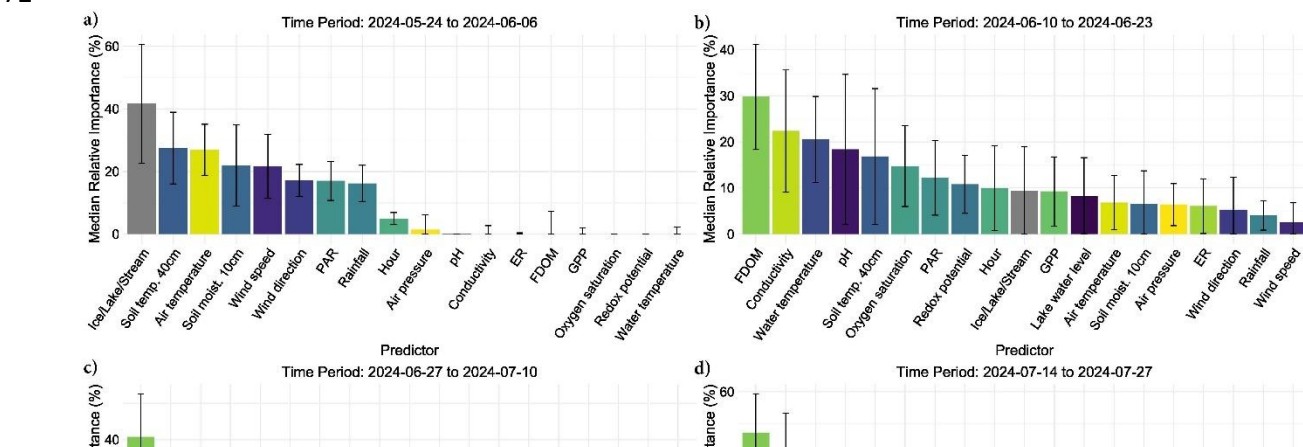


Figure 6. The figure illustrates the relative importance of environmental conditions predicting
diffusive CH$_4$ fluxes using bar-plots and standard error bars. Each predictor variable is on the x-axis,
while its percent importance for its inclusion in a fitted BRT is given on the y-axis where the standard
error bars represent variability in importance based on presence or absence in a given model. Each
bar color represents a distinct environmental condition.

### 3.3.1 Spring Thaw Phase: Peak Fluxes Driven by Hydrological and Climatic Controls

The spring thaw phase marks a shift in catchment conditions, where a frozen landscape gives
way to thaw season and hydro-connectivity between land and water is strong (Figure 5c-d, g-h). In
the spring thaw phase, initial peak fluxes of CH$_4$ were primarily dependent on increasing rainfall,
changing wind conditions, warming air and soil temperatures, and increased soil moisture content,
while low dissolved organic matter (i.e., FDOM) indicated increased fluxes (Figure 6a-b and 7a-b).
Soil moisture was found to have a Pearson's r = 0.93 with lake water levels, suggesting the lake levels
are strongly connected to snowmelt and groundwater hydrology (Figure S11). The distribution of
nutrients on the island has been shown to be linked to snowmelt and hill slope topography
(Westergaard-Nielsen et al., 2020), which is likely playing a role during the early part of the season,
but especially later in the year as DOM, a proxy for nutrients, becomes the primary limiting factor in
predicting higher fluxes (Figure 7b-c) (Olid et al., 2021, 2022). The processes driving CH$_4$ fluxes from
water surfaces is likely two-fold.

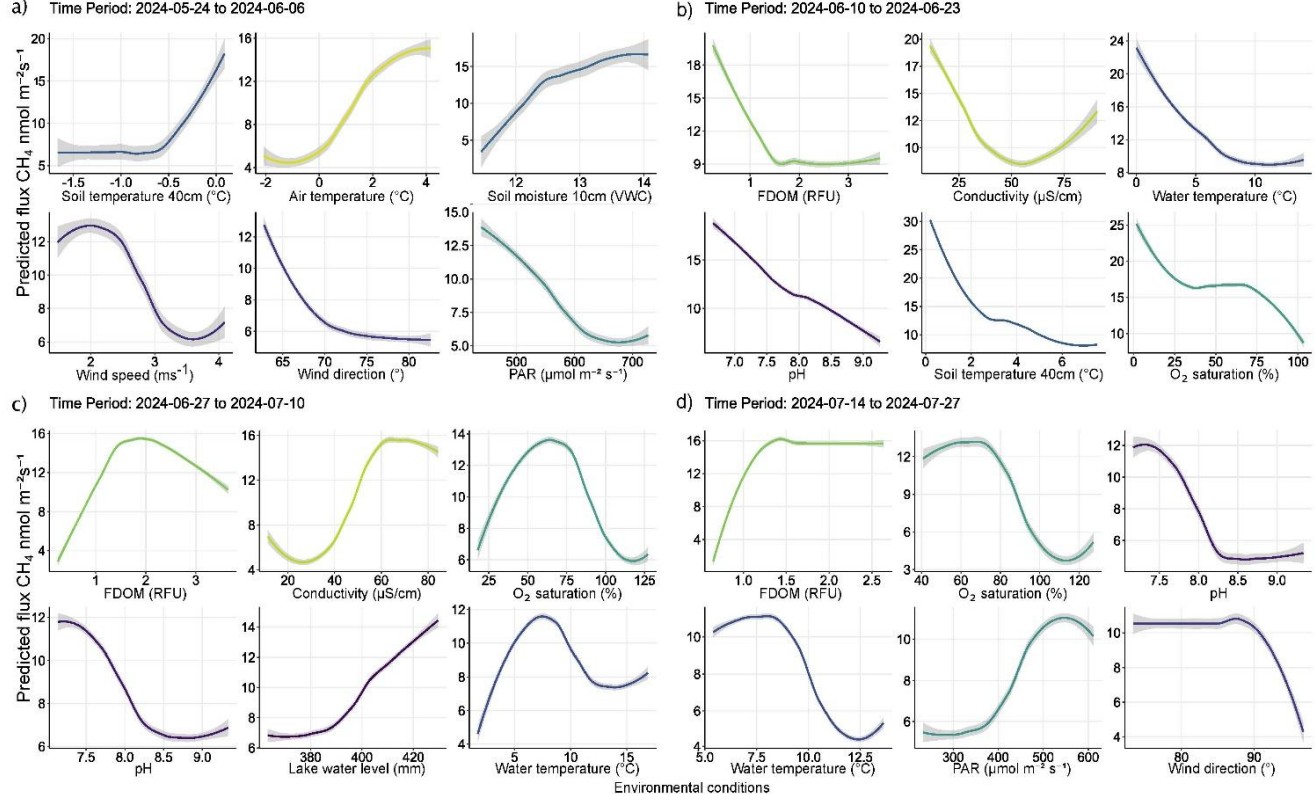


Figure 7. Partial dependency plots illustrating the indirect and direct marginal effects of various environmental conditions predicting diffusive CH₄ fluxes. Each panel displays the effect of a numerical predictor used during model training to predict CH₄ fluxes. Although some predictors may not have been directly involved in regulating fluxes from water surfaces, their evaluation reveals the relationship between water column conditions and catchment processes regulating CH₄ fluxes. The figures are ordered by importance in each time period and the colors correspond to those seen in Figure 6. The colored lines represent the result of a fitted general additive model (y ~ s(x)) and thus a smooth representation across the 321 fitted BRT models. The grey shaded area around the line represents ±SE (0.02 - 0.2). Each predictor and its numerical range are given on the x-axis, while predicted fluxes are given on the y-axis.

As snowfall turned to rain, the thawing of soils accelerated and water content in the active layer increased, potentially driving peak emissions via the lateral mobilization of dissolved CH₄ toward surface waters (Figure 7a-b) (Walter Anthony et al., 2012; Neumann et al., 2019; Olid et al., 2022).  However, as the thaw progressed, contributions to fluxes from catchment soils decreased as the upper layers began to dry and lake water levels reached their maximum (Figure 7b-c). DOM serves as a critical substrate for both CH₄ production and oxidation, particularly in permafrost-influenced regions where thawing can release large amounts of particulate matter (Keskitalo et al., 2021; Bouranis et al. 2025). Anaerobic conditions in water saturated soils and low oxygenated waters may have further driven fluxes during the thaw period as methanogenic microbial communities rapidly consumed incoming labile organic compounds (Neumann et al., 2019; Stevenson et al., 2021). However, as the thaw progressed, the lake briefly shifted to net-autotrophy (Figure S6), marking an important shift from hydroclimatic controls on fluxes, to a patchwork of biochemical transport pathways and barriers (Figure 7 b-d and Figure A1b-d). During this transition, increased GPP and the resulting oxygen saturation in the lake and streams enhance methanotrophic activity, thereby promoting methane oxidation and reducing net CH₄ emissions (Figure 7b–c and Figure A1b-c). The central role of oxygen availability and active methanotroph communities in

regulating methane fluxes during Arctic thaw has been demonstrated in both tundra landscapes and
permafrost mires (Christiansen et al., 2015; Graef et al., 2011; Singleton et al., 2018). As oxygen
production increases, greater amounts of oxygen become available to support methanotrophic
activity in the water column. Methanotrophic activity is likely represented in the observed reduction
of $CH_4$ fluxes during periods of decreased ER (i.e., when oxygen consumption is elevated, Figure A1c),
as higher oxygen availability enhances aerobic methane oxidation (Figure 7c). However, the streams
remain net heterotrophic despite increasing GPP and ER during this time period, therefore providing
a more suitable environment for $CH_4$ production and flux (Figure S6). Thus, the lake and streams
reflect distinct ecosystems for the production and flux of $CH_4$, where the lake being an overall
methane source, has a greater tendency to oxidize $CH_4$ following the thaw, while the streams follow
along a low-to-high flux gradient as the slope flattens and soils become saturated (Westergaard-
Nielsen et al., 2020; Stevenson et al., 2021).

### 3.3.2 Growing Season: Oxic-Anoxic Transport Pathways and Barriers

Early in the growing season the lake and streams enter into a transition phase, where peak
fluxes become increasingly dependent on biochemical pathways related to DOM availability,
conductivity, oxygen saturation and pH (Figure 7c). DOM is the limiting factor late in the season
where higher fluxes are increasingly dependent on water column conditions with greater levels of
DOM (Figure 6c-d and 7c-d). $CH_4$ fluxes during this time period were strongly associated with
indicators of microbial activity forming oxic-anoxic transport barriers, or pathways, respectively. For
example, microbial activity in anoxic sediments maybe producing $CH_4$, but whether it is diffused to
the atmosphere is directly affected by the micro-conditions of the water column (e.g., DOM,
conductivity, pH, and/or GPP/oxygen saturation), either forming an oxidative barrier, or an anoxic
pathway (Figure 7c-d and Figure A1). For example, groundwater transport of $CH_4$ from anoxic
sediments to oxygen-rich streams may result in water with high concentrations of both. While some
of the $CH_4$ is likely oxidized during transport, it can be that both are respired at turbulent sections of
the stream, which were the highest fluxes observed from the streams and during this phase of the
season (Street et al., 2016; Neumann et al., 2019; Olid et al., 2022; Kleber et al. 2025).
Fluxes are further affected by water conditions either favoring methanogenic, or
methanotrophic activity (Conrad, 2007; Cunada et al., 2021; Emerson et al., 2021). pH levels near
neutral likely indicate water conditions favorable to methanogenesis at the sediment-water
interface, while increasing alkalinity may reduce methanogenic and/or favor increased
methanotrophic activity as growing season progresses (Figure 7c-d). For example, during growing
season micro-pH and oxygen saturation conditions in the lake and streams are influenced by the
ever-increasing presence of macrophytes, mosses and plankton, which tend to drive pH and oxygen
levels higher (Liebner et al., 2011; Cunada et al., 2021). Here we show that increasing pH and oxygen
saturation, as a result of primary production, create an aerobic environment that favors
methanotrophic activity, thereby driving $CH_4$ emissions down through the growing season (Figure
7c-d, Figure A1). Declines in oxygen saturation driven by microbial respiration can create anoxic
conditions that enable $CH_4$ emissions from sediment to surface waters (Conrad, 2007; Michel et al.,
2010; Street et al., 2016; Cheng et al. 2024).  Such a mechanism likely explains the formation of flux
hotspots associated with decomposing iron-oxidizing bacterial mats along stream banks (Figure A2)
(Wallenius et al., 2021; Cheng et al. 2024).  In the case of the bacterial mats, we observed fluxes
were highest in the streams where bacterial assemblages had become exposed to the atmosphere
and were decomposing in stagnant water (Figure A2), which may suggest that the decomposition of
the bacteria was releasing dissolved organic substrates in a low-oxygen environment already primed
for methanogenic activity (Wallenius et al., 2021; Cheng et al. 2024). This idea is supported late in

the season when increased $CH_4$ emissions become dependent on niche environments where moderate levels of dissolved organic matter (FDOM) and low oxygenated water predict higher fluxes (Figure 7d and Figure A1d). However, submerged bacterial mats along with filamentous green algae the presence of macrophytes and/or mosses in the lake and streams, were associated with lower fluxes, suggesting they form oxidative barriers for $CH_4$ fluxes from the sediment when submerged (Figure 7d and Figure A1d) (Heilman & Carlton, 2001; Laanbroek, 2010; Liebner et al., 2011; Esposito et al., 2023). The submerged, or not status of bacterial assemblages' points to an interesting feedback mechanism between $CH_4$ fluxes and dropping water levels creating variability in emission pathways. Similar processes have been shown in relation to submerged brown mosses in Arctic tundra ecosystems have been shown to promote $CH_4$ oxidation and thus decrease $CH_4$ emissions from sediments (Žárský et al., 2018). Overall, these results highlight the broader importance of fine-scale biogeochemical dynamics shaping $CH_4$ fluxes from a permafrost catchment and provide an important data point in an uncertain region of the world.

## 4 Conclusion

This research provides a temporally resolved catchment scale $CH_4$ flux analysis across different waterbody types and conditions—lake, streams, and ice/snow-covered surfaces—subsequently describing important biogeochemical and climatic controls on emissions. Often lost in temporally coarse assessments is a detailed understanding of seasonal transitions in processes related to $CH_4$ fluxes and environmental control mechanisms. Leveraging BRT to fit hundreds of randomized models and visualizing the direct, and indirect controls on $CH_4$ fluxes reveals variability in how, for example, DOM and/or water temperature affect fluxes differently as the Arctic summer progresses. We presented an approach which captures ecosystem-scale effects, but furthermore describes isolated mechanistic effects related to, for example, GPP, revealing that primary productivity plays a critical role in regulating $CH_4$ emissions from permafrost affected waterbodies. This work contributes to understanding carbon feedback mechanisms in a region where process-level knowledge is needed to scale global models simulating $CH_4$ emissions from permafrost affected waterbodies.

## Code Availability

R code for calculating methane fluxes can be found here: https://github.com/mthayne527/fluxCH4.

## Data Availability

Meteorlogical can be accessed here: https://doi.org/10.17897/FEGK-0632, and soil data here: https://doi.org/10.17897/6G78-P793, https://doi.org/10.17897/9N7Z-GA63, and can be accessed via the Greenland Ecosystem Monitoring website: https://data.g-e-m.dk/datasets?theme=climate. BAWLD circum-Arctic waterbody dataset can be found here: DOI: 10.5194/essd-13-5151-2021. Water chemistry and chamber data can be requested from Torsten Sachs at Helmholtz Centre for Geosciences in Potsdam, Germany (GFZ).

## Author contributions

MWT collected, compiled and analyzed data, and wrote the manuscript. KK collected, compiled, and analyzed data, and contributed writing parts of the manuscript. CW provided methodological guidance and feedback, and contributed to writing parts of the methodology. AK provided comments, feedback, and guidance on interpreting results, and contributed to writing

various parts of the manuscript. TS collected data, provided comments, feedback, and guidance on interpreting results, and contributed to writing the manuscript.

## Competing interests

The authors declare that they have no conflict of interest.

## Acknowledgments

This research is part of the MOMENT project which is funded by the Federal Ministry of Research, Technology and Space (BMFTR) under grant number 03F0931E. We acknowledge the community of Qeqertarsuaq, Greenland for allowing us to research their land and water. We acknowledge the help received from all of the partners part of the MOMENT project, with a specific acknowledgment for Selina Undeutsch and Prof. Dr. Lars Kutzbach from the University of Hamburg. We acknowledge Dr. Evan Wilcox for collecting water level from the lake during the 2024 field season. We would acknowledge the University of Copenhagen and the Arctic Station team for providing an environment for conducting this research. We acknowledge the work of the Greenland Ecosystem Monitoring network and specifically Charlotte Sigsgaard for her help in getting meteorological and soil data. ChatGPT models 4.1 and 5 were used to edit parts of the manuscript.

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

# Appendix A

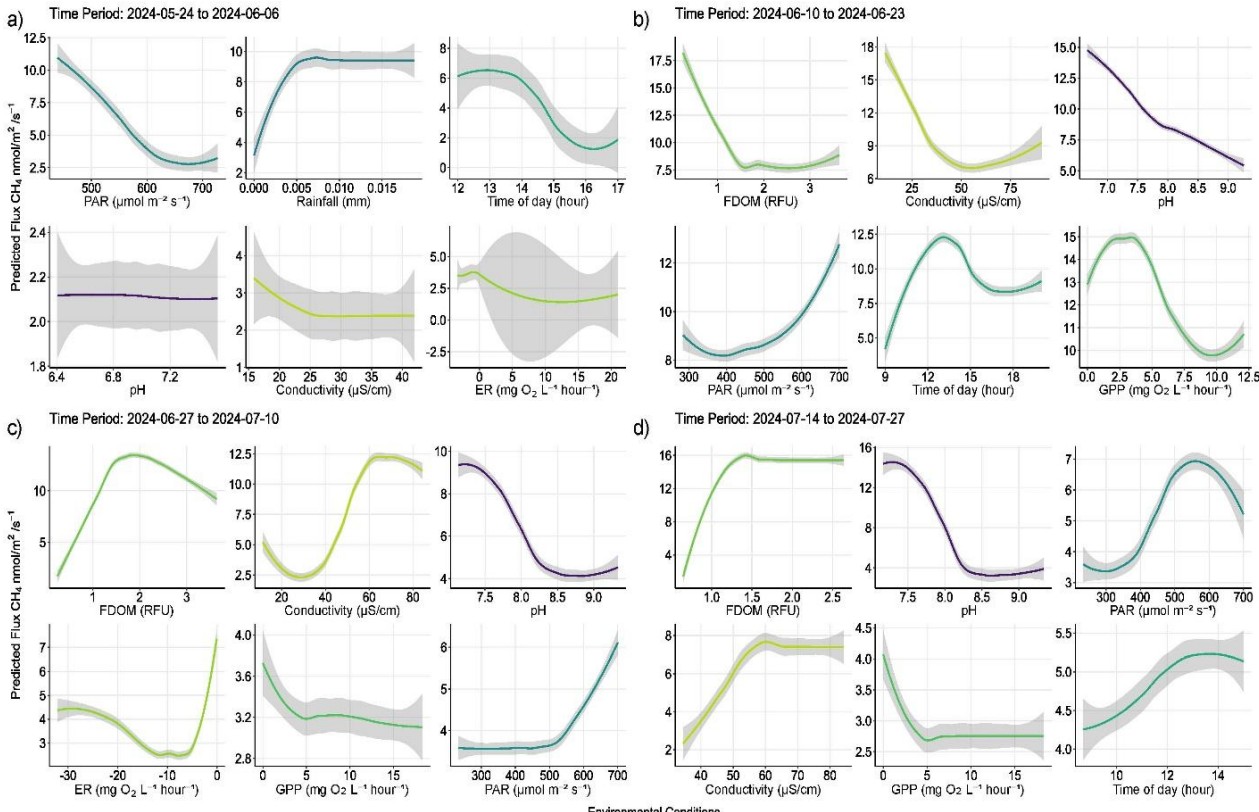

Figure A1. Partial dependency plots illustrating predicted marginal effects of meteorological, and biochemical water conditions predicting diffusive $CH_4$ fluxes. Each figure represents the direct marginal effect on $CH_4$ fluxes when all other predictors are held at their median, therefore giving a more mechanistic understanding of those conditions and processes regulating fluxes from water surfaces. The figures are ordered by importance in each time period and colors correspond to those seen in Figure 6 of the main text. The colored lines represent the result of a fitted general additive model ($y \sim s(x)$) and thus a smooth representation across the 321 fitted BRT models. Each predictor and its numerical range are given on the x-axis, while predicted fluxes are given on the y-axis.


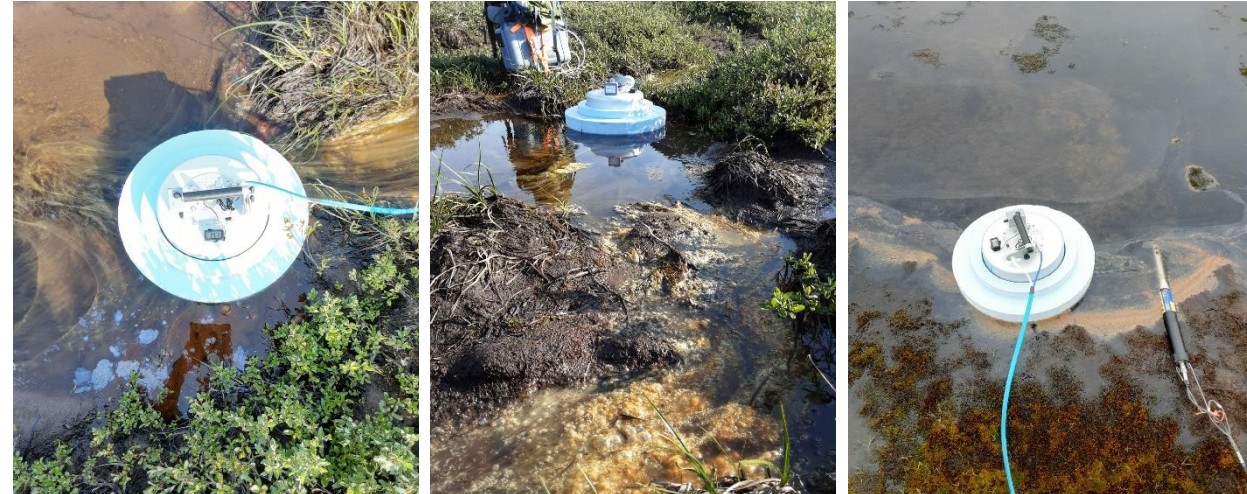

Figure A2. Illustrates the various microbial forms encountered and found to influence CH$_4$ fluxes
from the catchment area. In the photo on the far left we can see gas film on the water surface which
was associated with submerged iron-oxidizing microbial assemblages, i.e. similar to what is shown in
the middle photo, however exposed to the atmosphere in lower water levels. The photo on the right
shows a brown alga which formed in the warm spring area southeast of the lake. In all cases,
increased fluxes were generally encountered when measuring atop the middle and right microbial
assemblages.