# Peer review of "Spatiotemporal Variability and Environmental Controls on Aquatic Methane Emissions in an Arctic"

_EGUsphere, 2025_

## Author Response (AR2)

We would like to thank the editor for allowing us to clarify these points and submit a revised manuscript. We conducted additional post-hoc testing and modelling to address the important comments made by the editor. We believe the additional analysis and clarifications justify our conclusions.

**Editor Comment:** The self-built chamber was made of semi-transparent plastics. Note that some commonly used plastics, such as polyethylene, can produce methane under solar radiation (e.g., Royer et al., 2018, https://doi.org/10.1371/journal.pone.0200574). Were any tests conducted to check the blank of the chamber? The identical median fluxes between 2023 and 2024 do not necessarily warranty the comparability of the two different chambers unless the factors controlling methane fluxes were identical (both qualitatively and quantitatively) between the two years, which is doubtful.

**Author Response:** This is an important point, and we acknowledge that potential chamber-derived $CH_4$ production was not explicitly considered in our initial analysis. Based on Royer et al. (2018), trace methane production from prolonged exposure to photochemical reactions involving plastic components is plausible under warm and extended UV exposure. To assess whether the change in chamber construction between years introduced systematic bias in $CH_4$ flux estimates, we constructed a linear model using log-transformed flux from the two years as the response and chamber type, spatial coordinates (latitude, longitude), and Julian day as predictors. Chamber type was not a significant predictor ($p = 0.136$), and a QQ plot comparing log-transformed fluxes across years showed no systematic deviation across the flux distribution, particularly in the central or upper range. These findings—alongside identical median fluxes and consistent spatial patterns between years—support our conclusion that chamber design did not meaningfully bias the measurements. It is also important to note that our field deployments occurred under cool Arctic conditions, and the chambers were neither exposed to prolonged sunlight nor exposed to extreme prolonged photodegradation scenarios described by Royer et al. (2018). Please **see lines 132-138** and Figure S1:

"However, to evaluate whether changes in chamber construction between years introduced systematic bias in $CH_4$ flux estimates, we fit a linear model using log-transformed flux as the response and chamber type, latitude, longitude, and Julian day as predictors. Chamber type was not a significant predictor ($p = 0.13$), and QQ plots of log-transformed fluxes across years showed no consistent deviation across the flux distribution, except at lower emission rates (Figure S1). Together, these findings suggest that differences in chamber construction did not substantively influence the observed fluxes."

**Editor Comment:** It's impressive that the authors measured/collected numerous parameters as potential predictors of methane fluxes. However, I did not see they included water depths at each flux measurement spot (I apologize if I missed this info). Given that the lake was very shallow (<4.5 m) and that many of the flux measurements were located within 0.2 m from the shore, sedimentary release of methane into the water column likely contributed substantially to the methane fluxes determined. Please explain why water depth was not included as a potential predictor.

**Author Response:** We thank the editor for this thoughtful comment. While we did record water depth at each chamber deployment using the AquaTroll 600, we did not include it as a predictor in the BRT models. In shallow Arctic lakes such as Sanningasup Tasia (<4.5 m), methane emissions originate primarily from sediments across the depth gradient. However, increasing water depth often enhances methane oxidation potential via elevated oxygen exposure and microbial activity (Bogard et al., 2014; Emerson et al., 2021; Bulínová et al., 2025). As such, we prioritized direct chemical and biological predictors—such as dissolved oxygen saturation, FDOM, and GPP—that

more precisely reflect the processes driving $CH_4$ emissions (Christiansen et al., 2015; Singleton et al., 2018). Additionally, our spatial sampling explicitly accounted for depth variability: we deployed chambers both along the shallow shoreline (<0.2 m) and across the lake interior (~3–4.5 m) using a boat. This ensured that depth-related variation in flux and water chemistry was embedded in the dataset, reducing the need to rely on water depth as a proxy. We now clarify this rationale in the methods **on lines 298-307 and with Figure S10**:

"Although water depth was recorded during chamber deployments using the AquaTroll 600, it was not included as a predictor in the BRT models. In shallow Arctic lakes like Sanningasup Tasia (<4.5 m), $CH_4$ is primarily sediment-derived, with deeper zones more likely to promote oxidation or dilution due to greater oxygen exposure (Bogard et al., 2014; Bulínová et al., 2025; Emerson et al., 2021; Li et al., 2020). Shoreline fluxes often dominate due to anoxic, vegetated sediments, while interior zones tend to suppress emissions (Thompson et al., 2016; Kyzivat et al., 2022; Rasilo et al., 2015). We therefore prioritized biogeochemical water column predictors—FDOM, dissolved oxygen saturation, and GPP—over depth (Christiansen et al., 2015; Singleton et al., 2018), and explicitly captured depth gradients via boat-based chamber deployments across the lake interior."

**Additional technical corrections suggested by the editor:**

**Editor Comment:** What does the violet color represent in Figure 5? The color bars do not contain this color.

**Author Response:** This is a good point and we acknowledge this in the figure caption on lines 426-427, but unfortunately did not point out the color associated with the sentence. We edited the figure caption and added clarification **on lines 426-427**:

"Areas outside of the water were not measured and are artifacts from the KDE mapping algorithm *interacting with the image (i.e., purple color = no data)*."

**Editor Comment:** Figure 6 caption: Delete "The figure illustrates".

**Author Response:** Thanks for the suggestion. We replaced the suggested text **on line 475** with:

 "Bar charts showing the relative importance of…"

We additionally changed text with similar framing in the figure caption of A2. **See line 905.**

**"***Image showing the* various microbial forms…"

**Editor Comment:**  Figure 7: Y-axis title: Change to "Predicted CH4 flux" or "Predicted flux of CH4".

**Author Response:** This is a good point. We have changed the y-axis title for both figure 7 (**line 493**) and figure A1 (**line 886**) to "Predicted $CH_4$ flux (nmol m$^{-2}$s$^{-1}$)"

**Editor Comment:**  Line 603 (Author contributions): "contributed to"

**Author Response:** We have added "contributed to" directly after each co-author initials. **See lines 603-608**

**Editor Comment:**  Line 614 (Acknowledgments): delete "part" after "partners" or change "partners" to " partners' ".

**Author Response:** Thank you for catching this. We deleted "part" from the sentence. **See line 615.**

**Editor Comment:**  References: several doi citations miss "https://". Please check.

**Author Response:** Thanks for pointing this out. We looked more in detail at the formatting of references and they should now be formatted correctly for the journal. Please **see lines 623-883.**

**Editor Comment:** Please carefully check, and correct if needed, spelling, grammar, sentence structures, unit formats throughout the manuscript and its supplementary materials (including figures and tables).

**Author Response:** We would like to thank the editor for the opportunity to publish in Biogeosciences and taking the time to improve and clarify our manuscript before publication. We read through the manuscript and believe we have made all the appropriate edits and clarifications.

We thank the reviewer for the thoughtful and constructive feedback provided on our manuscript entitled *"Spatiotemporal Variability and Environmental Controls on Aquatic Methane Emissions in an Arctic Permafrost Catchment."* We have carefully addressed all comments, revising the manuscript where appropriate and providing clarifications where necessary. Below, we provide detailed, line-by-line responses with the full reviewer comments included. Each Reviewer Comment is reproduced in full, followed by our Response, including specific changes made to the manuscript (with line numbers referring to the revised manuscript).

**Line 25 (Fox-Kemper et al., 2021)**
**Reviewer Comment:** "Line 25 (Fox-Kemper et al., 2021)."
**Response:** We thank the reviewer for catching this. A missing parenthesis was added to the reference citation on **line 27**.

**Line 40 (Saunois et al., 2016 → 2025)**
**Reviewer Comment:** "The reference to 'Saunois et al., 2016' can be updated in 2025."
**Response:** We thank the reviewer for pointing this out and for noting the updated reference. The citation has been updated to Saunois et al. (2025) on **line 41**.

**Line 64 — Sentence rephrasing for readability**
**Reviewer Comment:** "Methanotroph and methanogen microbial assemblages along an upland-wetland environmental gradient were…" I suggest rephrasing this sentence to improve readability.
**Response:** We thank the reviewer for this excellent suggestion. The sentence has been rephrased for clarity and parallel structure on **lines 64-67**:

"Sedimentary processes in lakes promote carbon storage, whereas methanotrophic and methanogenic microbial assemblages along an upland–wetland environmental gradient regulate $CH_4$ consumption and emission, respectively."

**Line 104 — Figure 1 clarity**
**Reviewer Comment:** "I suggest outlining the specific locations of wetlands, rivers, and tributaries in Fig. 1."
**Response:** We appreciate this helpful recommendation. We have revised the Figure 1 to highlight these areas and the caption was also updated to specify spatial features on **lines 112-117**:

"The blue circle indicates streams fed by snowmelt and the turquoise circle indicates the outlet of the lake. The red circle indicates the warm spring area which forms a wetland type ecosystem. Points inside the black circle north of the lake indicate measurements taken from the Red River and its stream tributaries."

**Line 121 — Chamber material and sealing performance**
**Reviewer Comment:** "Semi-transparent plastic material. This is one of my concerns in the paper. Could you please give detailed information about this chamber, e.g., sealing performance? This raises my concern because the plastic material is not a 'regular' material in detecting trace gases. This may cause systematic error when calculating methane fluxes by using linear fitting. In addition, the data between 2023 and 2024 were collected by using different chamber types, which weakens the results and discussion when comparing these two years of data."
**Response:** We thank the reviewer for this important observation. We have added a detailed clarification in Section 2.2 on **lines 128-138**:

"The semi-transparent plastic chamber used a circular foam floater that was wrapped around the outside of the chamber, allowing 2cm of the chamber to be below the water surface, forming a

100% airtight seal. The West Systems chamber was inserted into a closed-cell foam floater, where the seal was created once the chamber was inserted into the floater. Despite differing chamber materials and flotation devices, median fluxes between 2023 and 2024 were identical at 5.0 nmol m$^{-2}$s$^{-1}$. However, to evaluate whether changes in chamber construction between years introduced systematic bias in CH$_4$ flux estimates, we fit a linear model using log-transformed flux as the response and chamber type, latitude, longitude, and Julian day as predictors. Chamber type was not a significant predictor (p = 0.13), and QQ plots of log-transformed fluxes across years showed no consistent deviation across the flux distribution, except at lower emission rates (Figure S1). These findings suggest that differences in chamber construction did not substantively influence the calculated fluxes."

**Wind speed and flux uncertainty**

**Reviewer Comment:** "Another concern is that the calculation fluxes were not revised for the real wind speed. I know this is difficult based on current data, but a discussion on how the wind speed would disturb the water surface and methane emission should be included. Besides, an uncertainty evaluation of chamber-based flux should also be included so that it is convenient to compare with the fluxes reported in other areas."

**Response:** We thank the reviewer for these valuable points. The following additions have been made:

*Section 2.2 (Data Collection**) lines 155-159**:*

"While floating chambers isolate the headspace from light wind disturbance, increased surface turbulence may influence gas exchange in the open space of the chamber (Vachon and Prairie 2013). Our approach captures diffusive exchange under mostly calm-water conditions (i.e., wind speed up to ~4 ms$^{-1}$), but we acknowledge that regional wind-driven mixing may contribute to flux variability beyond our individual chamber footprints."

*Section 2.3 (Flux Algorithm & Ebullition Detection) **lines 223-224**:*

"Because fluxes were derived from high-frequency (1 Hz) concentration data fitted using a GAM with a strict acceptance threshold of R$^2$ ≥ 0.99, analytical uncertainty in the rate-of-change estimation is negligible. Conventional uncertainty propagation (e.g., based on regression slope error or replicate chambers) is not meaningful in this context because the GAM approach fits a smooth curve through hundreds of data points per deployment, effectively minimizing noise and preventing poor-quality fits from contributing to the final flux values. This ensures that the dominant source of variability in the dataset reflects true environmental heterogeneity rather than analytical error. Furthermore, because the chamber headspace was fully sealed and isolated from external turbulence, wind-induced variability—which often motivates uncertainty corrections—is mechanically removed from the flux calculation process. For these reasons, we report spatial variability (e.g., medians, ranges, and interquartile spread) rather than analytical uncertainty, as it provides a more ecologically relevant representation of flux variability across the catchment."

**Line 138 — Frequency of chamber measurements**

**Reviewer Comment:** "Did the 707 chamber measurements take place twice (2023 and 2024) or several times on different days? I think this information is important to evaluate the significance of this study, though I found some of it in the figures."

**Response:** We thank the reviewer for this question. The following clarification was added to Section 2.2 on **line 152-155**

"Overall, we made 707 chamber measurements, *representing ~10-15 chamber placements taking place daily,…*"

**Line 197 — Typographical error**
**Reviewer Comment:** "Typo after the word 'ebullition.'"
**Response:** We thank the reviewer for spotting this. The typographical error has been corrected and clarified as versus instead of v., on **line 230**.

**Line 205 — Dissolved oxygen (DO) concentration method**
**Reviewer Comment:** "How did you measure the DO concentration? At a one-minute frequency, I suppose a probe was used. The method, precision, and uncertainty should be clarified."
**Response:** We thank the reviewer for the helpful suggestion. The following details have been added to Section 2.5:

*Section 2.2 (Data Collection***) lines 161-162***:

"…water parameters were collected in 2024 using a Aquatroll 600 water sonde (see section: Decoding Methane Drivers)."

*Section 2.4 (Lake and Stream Metabolism***) lines 240-241***:

"…(DO) collected at a one-minute frequency during chamber measurements (DO sensor accuracy: ± 0.1 mg L$^{-1}$; resolution 0.01 mg L$^{-1}$)."

**Line 223 & 244 — Section numbering correction**
**Reviewer Comment:** "Should be section 2.6 instead of 2.4" and "Should be section 2.7."
**Response:** We thank the reviewer for catching these numbering inconsistencies. They have been corrected in the revised manuscript.

**Line 256 — Soil temperature measurement depth**
**Reviewer Comment:** "Why did you use the soil at 40 cm but not at the surface or top 10 cm to match the soil volumetric water content's standard?"
**Response:** We thank the reviewer for this insightful comment. The following clarifications have been added on **lines 309-311:**

"We used catchment soil temperature at 40 cm to represent subsurface active-layer conditions that influence deeper thermal dynamics, groundwater inflow, and delayed soil heat retention through the thaw season."

**lines 314-316**

"Additionally, we used VWC at 10 cm depth because it was the most complete and continuous dataset across the measurement depths, and highly correlated with VWC at 20cm, 30cm and 40cm."

**Line 270 — Predictors count inconsistency**
**Reviewer Comment:** "'7 of the 21 weighted predictors.' This was inconsistent with that in Fig. 3, which says 8 of the 21 of the weighted predictors. Please check."
**Response:** We thank the reviewer for noticing this inconsistency. The text in Fig 3. has been corrected to "7 of the 21 weighted predictors **on lines 362-367**.

**Line 335–338 — KDE explanation placement**

**Reviewer Comment:** "The discussion jumped from different figures, weakening the flow. In addition, these sentences were saying the advantages of KDE, which I think is not suitable in the Discussion, but in the Method."

**Response:** We thank the reviewer for this valuable recommendation. The KDE methodological explanation has been moved to Section 2.4 (Spatial Flux Evaluation, **lines 263-265**), leaving only interpretive references in the Discussion.

**Additional Clarifications:**

- Clarified that all BRT models were trained using 2024 data only on **lines 285-286**.

"Considering we only collected water temperature in 2023, we used flux data from 2024 for training the BRT."

**Final Note:**

We sincerely thank Reviewer 1 for the thorough and constructive comments. These suggestions have substantially improved the clarity, rigor, and overall presentation of the manuscript.

We thank the reviewer for the thoughtful and constructive feedback on our manuscript *"Spatiotemporal Variability and Environmental Controls on Aquatic Methane Emissions in an Arctic Permafrost Catchment."* We have carefully considered all comments and revised the manuscript accordingly, providing clarifications or edits where necessary. Below, we present a detailed, point-by-point response. Each Reviewer Comment is reproduced in full, followed by our Response, including specific changes made to the manuscript (with line numbers referring to the revised manuscript).

**General Comment — Overall Study and Conclusions**
**Reviewer Comment:** "The authors deployed floating chambers during 2023 and 2024 summer to measure $CH_4$ amount and applied General additive model (GAM) plus binary regression tree (BRT) to fit $CH_4$ fluxes and separate ebullition from diffusion in a small arctic permafrost catchment. Further analysis of $CH_4$ fluxes against dynamics of other environmental variables highlighted a dynamic and intertwined network of environmental variables affecting each other, and determined the spatial and temporal variations in $CH_4$ fluxes as the outcome. The major conclusion and explanation is valid, though I have questions on some of the details."

**Response:** We thank the reviewer for this positive overall assessment. We are pleased that the major conclusions were found to be valid. The detailed questions and concerns are addressed in our responses below, and we have revised the manuscript to clarify all the specific points raised.

**Ebullition Terminology — Usage in Abstract and Main Text**
**Reviewer Comment:** "I have one major concern on 'ebullition'. Based on the paper description, I did not find authors using other data sources to validate the ebullition detection. Authors explained that the 'ebullition' in their study is actually 'non-linear concentration increases', which I believe include ebullition events (since quasi-ebullition is steady and can be similar to diffusive fluxes) and sudden increase of diffusive fluxes. I would strongly suggest authors to reconsider using 'ebullition' in the main text, especially abstract, which is a well-defined gas transport pathway."

**Response:** We appreciate the reviewer's concern regarding our use of the term "ebullition." We agree that our algorithm identifies non-linear concentration increases that are *indicative* of ebullition events, but were not independently confirmed in every case (e.g., by direct observations). To avoid misunderstanding, we have revised the wording in the Abstract and throughout the text to clarify this. In the Abstract, rather than stating "Diffusive and ebullitive fluxes were derived…", we now write on lines 14-17:

> "Diffusive fluxes dominated (~98% of observations), while only ~1% of chamber deployments exhibited non-linear concentration increases indicative of ebullition, while the other ~1% were attributed to uptake."

This revised sentence removes the implication that we directly measured classical ebullition and clearly frames those events as *"indicative of ebullition"*. We have similarly adjusted terminology in the main text. For example, in **Section 2.3 (Flux Algorithm & Ebullition Detection)** we added a clarifying statement on **lines 234-236**

""Ebullition" in the context of the algorithm refers to a sudden, non-linear $CH_4$ increase identified by the algorithm, which includes observed ebullitive events, but does not strictly infer all fluxes calculated this way were from bubbles entering the chamber."

We now also place the term in quotes (**on lines 269 and 287**) when describing these occurrences.

**Line 19 — Timing of Shift to Biogeochemical Controls**
**Reviewer Comment:** "Line 19: 'later shifting to biogeochemical controls', how late? Several days or several weeks? A precise number can be the best."

**Response:** We agree that the timing of the shift to biogeochemical controls should be specified. In the Abstract, we have added an approximate timeframe to clarify what *"later"* refers to. The sentence now reads (18-21):

"Model results suggest that thaw-season $CH_4$ fluxes were initially driven by meteorological conditions and catchment soil conditions, but shifted rapidly—within approximately one week after ice-off—to biogeochemical controls, including dissolved organic matter, oxygen saturation, and pH."

**Line 28 — Microbial Production of Carbon (Rephrasing)**
**Reviewer Comment:** "Line 28: 'which produce carbon as a result of their metabolic processes' shall be 'which accelerate decomposition of soil organic carbon as a result of their metabolic processes' or 'which increase portable carbon input to atmosphere and riverine system as a result of their metabolic processes'."

**Response:** We appreciate these suggestions for clarifying the role of microbial metabolism. We have revised the sentence in the Introduction to more accurately describe the process. The phrase has been changed **on lines 29-30**:

> "…which accelerate the decomposition of soil organic carbon as a result of their metabolic processes".

This wording makes it clear that microbes are breaking down organic carbon, thereby releasing carbon (in gaseous or dissolved form) rather than literally creating new carbon.

**Line 66 — Redundant "systems" in Text**
**Reviewer Comment:** "Line 66: Shall remove 'systems' after 'freshwater ecosystems'."

**Response:** We have removed the extra word *"systems"* so that the sentence now reads **on lines 66-67**: *"Therefore, freshwater ecosystems..."*

**Figure 1 — Figure Quality and Compass Rose**
**Reviewer Comment:** "Figure 1: Can authors improve the figure quality? Could it be better if I use pictures with DPI >= 300? Also, it is better to add a simple compass rose to the map."

**Response:** We have improved Figure 1 as suggested and further labeled more clearly the different waterbodies. The figure has been regenerated at high resolution (at least 300 DPI) to ensure clarity in print and digital formats. Additionally, we have added a north arrow (compass rose) to the map. See new figure and caption **on lines 112-117**.

**Line 121 — Photograph of the Self-Built Chamber**
**Reviewer Comment:** "Line 121: 'We used a self-built cylindrical chamber made of semi-transparent plastic' I can barely imagine what your chamber looks like, but it will be very helpful if you don't mind uploading a picture of your equipment, since it is self-built. This is more like a suggestion rather than request."

**Response:** We appreciate the reviewer's interest in the chamber design. In response, we have included a photograph here in the point by point of the self-built chamber used in 2023.

[Figure]

However, we have left it out of the manuscript as the Boosted Regression analysis and determination of emission drivers were based on data collected using the industry built West Systems Chamber seen in Figure A2 of the appendix. However, we make clear that despite differing construction the median fluxes between the years were identical at 5.0 nmol $m^{-2}s^{-1}$ and further show no systematic bias resulting from chamber construction. We clarify this point **on lines 128-138**.

**Line 135–138 — Rationale for Control Period Measurement Change**
**Reviewer Comment:** "Line 135 ~ 138: What is the major reason for changing the measuring object for the control period?"

**Response:** The reviewer is correct that our approach to "control" measurements differed between 2023 and 2024. We now explicitly clarify our rationale **on lines 149-153**:

> "In 2023, we measured isolated meltwater pools during the thaw to represent control conditions (water not yet connected to the lake or streams). In 2024, to capture an even earlier baseline, we conducted chamber measurements on top of snow and lake ice prior to thaw onset, providing a true pre-thaw control period."

**Line 173 — GAM Application for 2023 vs 2024 Data**
**Reviewer Comment:** "Line 173: Since you have certain different treatment on equipment between 2023 and 2024, are you fitting a general GAM for both years or one for each year separately?"

**Response:** We thank the reviewer for this comment. As stated **on lines 161-163** in Section 2.3 of the manuscript, the same GAM-based flux calculation algorithm was applied uniformly to individual chamber measurements. However, to clarify we controlled for the different chamber construction we added the following text **on lines 169-171**:

"The flux calculation procedure was applied identically to each chamber time series for both 2023 and 2024, while controlling for different chamber construction."

In addition, we clarify **on lines 194-195**:

"…the *concentration time series* is then fit to a GAM (i.e. gam(CH4$_{moles}$ ~ s(time, k = gam_knots))…"

**Line 202 & 223 — Section Numbering Corrections**
**Reviewer Comment:** *"Line 202: Shall be section 2.4"* and *"Line 223: Shall be section 2.5"*

**Response:** Thank you for catching these section numbering errors. We have corrected the section headings.

**Line 203 — References & Validation for Metabolic Flux Estimates**
**Reviewer Comment:** "Line 203: There're a bunch of assumptions you used for estimating different metabolic fluxes, can you put some reference or available 3rd party data to validate your results?"

**Response:** We thank the reviewer for raising this important point. The methods used to estimate NEP, GPP, and ER closely follow established and widely adopted protocols and DO assumptions from previous studies (e.g., Garcia & Gordon 1992; Cole & Caraco 1998; Mulholland et al. 2001; Winslow et al. 2016; Hall & Madinger 2018). However, we now clarify the uncertainty related to DO measurements **on lines 240-241**:

"…(DO) collected at a one-minute frequency during chamber measurements (DO sensor accuracy: $\pm$ 0.1 mg L$^{-1}$; resolution 0.01 mg L$^{-1}$)"

Furthermore, we have added more references which specifically estimated NEP, GPP, and ER from Arctic and Boreal lakes and streams **on lines 255-258**:

"Nonetheless, the approach captures broadly the metabolic trends in lake and stream metabolism observed in other Arctic and Boreal waterbodies (Mulholland et al., 2001; Rocher-Ros et al., 2021; Ayala-Borda et al., 2024; Klaus et al., 2022; Myrstener et al., 2021) and is useful for comparing fluxes across aquatic biomes."

Our NEP, GPP, and ER results fall well within the values reported in the cited literature.

**Line 220 — Data Smoothing and Temporal Resolution**
**Reviewer Comment:** "Line 220: This also depends on how you can smooth your collected data to a coarser temporal resolution."

**Response:** We thank the reviewer for this comment. To clarify, DO data used in our metabolism calculations were collected natively at one-minute intervals using a water sonde see added DO uncertainty **on lines 240-241**. No additional smoothing or aggregation was applied.

**Line 236 — Typographical Error in Statistical Test Name**
**Reviewer Comment:** "Line 236: 'Kurskal-Wallis test' shall be 'Kruskal-Wallis test'?"

**Response:** Thanks for catching this. We have corrected "Kurskal-Wallis" to "Kruskal-Wallis" in the text **on line 274.**

**Line 250–253 — Measurement of Environmental Variables (Instrumentation)**
**Reviewer Comment:** "Line 250 ~ 253: I'm not sure how you obtained these measurements? What equipment did you use to measure these variables?"

**Response:** Thank you for your attention to this detail. We state **on lines 159-165** in *Section 2.2 (Data Collection),* which device was used and/or where the water, meteorological, and soil parameters come from. We state:

"We simultaneously measured water temperature using Truebner EC-100 RS-485 EC/Temperature sensors in 2023 and a suite of water parameters were collected in 2024 using an AquaTroll 600 water sonde (see section: Decoding Methane Drivers). Meteorological data and soil characteristics were collected from nearby meteorological stations maintained by Aarhus University which are part of the Greenland Ecosystem Monitoring Database (Greenland Ecosystem Monitoring, 2025a-d) (see section "Decoding Methane Drivers" for list of variables used)."

**Line 256 & Fig. S7 — Soil Temperature Depth and Relevance to Aquatic Fluxes**

**Reviewer Comment:** "Line 256 & Fig. S7: How do you account for the soil temperature depth for stream and lake? Start from the bottom of the lake? In line 256 you mentioned using soil

temperatures 'at 40 cm' to analyze correlation to surface water CH₄ fluxes, which seems to be quite irrelevant if it means 40 cm below lake bottom."

**Response:** Thank you for the question. Please see the next response as we have combined this comment with the next as they are addressing the same issue.

**Line 245–267 — Clarification of Variables Measured for Lake vs Stream vs Upland**
**Reviewer Comment:** "Line 245 ~ 267: This paragraph shall clarify what different variables you measured for lake, stream and upland separately. If all variables are measured, shall explain if any different treatment is applied. For example, soil volumetric content seems not to be a measurable variable for lakes?"

**Response:** We apologize for the confusion here and thank you for pointing this out – the soil temperature at 40 cm is measured in terrestrial soil within the catchment at a nearby climate station, not under the lake or stream bed. Soil characteristics, temperature and VWC, were included to gain an understanding of how catchment hydrogeological conditions indirectly effect fluxes from surface waterbodies within the catchment. We have clarified where the suite of predictors come from, and why they were included in the models to ensure there is no misunderstanding **on lines 284-298 and 307-316**:

"To characterize fluxes we assembled a comprehensive set of predictors including; (a) aquatic variables measured in-situ with a water sonde at each chamber deployment (e.g., Conductivity (µS/cm), pH, redox potential (mV), dissolved oxygen (mg/L), oxygen saturation (%), water temperature (°C), and fluorescent dissolved organic matter (FDOM; RFU)) (Figure S6), (b) catchment soil characteristics collected at nearby climate stations (e.g., soil volumetric water content at 10 cm and soil temperature at 40 cm), and (c) meteorological variables collected from a nearby climate station (e.g., Surface air temperature at 2 cm (°C), Air temperature (°C), relative humidity (%), air pressure (mbar), precipitation (mm), PAR (µmol m⁻²s⁻¹), and mean wind speed (ms⁻¹) and direction (°)) (Figure S6). Lake water levels (mm) were included to characterize the effect of changing hydrologic conditions and its influence on lake CH4 fluxes... ...Catchment soil characteristics were included to capture the hydrogeological conditions surrounding the catchment. We used catchment soil temperature at 40 cm to represent subsurface active-layer conditions that influence deeper thermal dynamics, groundwater inflow, and delayed soil heat retention through the thaw season. Soil volumetric water content (VWC) at 10 cm was included to gain an understanding if dryer, or wetter soil catchment conditions effect surface water CH₄ fluxes, and to act as a substitute for water level in the lake early in the season as these two share a Pearson's correlation of r = 0.93. Additionally, we used VWC at 10 cm depth because it was the most complete and continuous dataset across the measurement depths, and highly correlated with VWC at 20 cm, 30 cm and 40 cm."

In short, we have made it clear that the soil temperature at 40 cm is not measured beneath the lake or stream, but rather in an adjacent terrestrial environment, and we justified its inclusion for capturing broader catchment-wide hydrogeological effects.

**Line 277 — Choice of a Low Bag Fraction in BRT**
**Reviewer Comment:** "Line 277: A more popular range of bag fraction is 0.5 ~ 0.8, while here authors chose a relatively low bag fraction, it might make sense that authors expect more smoothed time series rather than over-fitting since the measured methane might be quite non-linear. Suggest explaining your choice."

**Response:** The reviewer is correct that our chosen bag fraction for the BRT models was on the lower end (we tested 0.30, 0.40, 0.50, as noted in Section 2.6). We have added an explanation of why we chose these bag fractions **on lines 329-333**:

 "While bag fraction values in the range of 0.5–0.8 are more commonly used, a lower bag fraction increases stochasticity in tree construction, which helps reduce overfitting—especially important for modeling noisy and highly non-linear $CH_4$ flux data. This conservative approach favors identifying robust general patterns rather than fitting noise or outliers."

**Figure 4 — Relating Emission Classes to Map (Figure 5)**
**Reviewer Comment:** "Fig. 4: Is it possible to show the relative locations of different lake and stream classes in Fig. 5?"

**Response:** This is a good point, also brought up by the other reviewer, and rather than linking Figure 4 and 5, we added some clarifications to figure 1 to give a better idea of where the different water body classes are. **See lines 112-119**.

**Line 286 — Calculation of Deviance in BRT Models**
**Reviewer Comment:** "Line 286: How did you calculate deviance?"

**Response:** Good point. We have added clarification of this calculation **on lines 341-345**:

"We further calculated the percent deviance explained for each fitted BRT model using the formula: % deviance explained = 100 × ((null deviance − residual deviance) / null deviance), where the null deviance represents the deviance of a model using only the mean response, and the residual deviance is from the fitted BRT model."

**Figure 7 — Explanation of Shaded Areas in Partial Dependence Plots**
**Reviewer Comment:** "Figure 7. I assume the shaded area represents standard deviation, but authors shall add explanation."

**Response:** We have updated the Figure 7 caption to explain the meaning of the shaded areas in the partial dependence plots. The caption now includes a sentence **on lines 498-499**:

 "The grey shaded area around the line represents ±SE (0.02 - 0.2)."

**Line 389 — Rephrasing Sentence about Flux Dissipation Through Season**
**Reviewer Comment:** "Line 389: 'Fluxes dissipated through the season until fluxes were isolated to the warm spring inlet and the eastern inlets (Figure 5).' I find it hard to interpret this sentence. Can you rephrase it?"

**Response:** We have rephrased the sentence in the **Discussion** to make the meaning clearer. The original intent was to convey that as the summer progressed, methane emissions decreased everywhere except in a couple of specific locations (the warm spring inlet and the eastern inlets to the lake, which continued to emit $CH_4$). See revised sentence **on lines 445-447**:

 "As the summer season progressed, $CH_4$ fluxes declined across most of the catchment, becoming largely confined to the warm spring inlet and the eastern inlet streams (Figure 5)."

**Line 429 — DOM vs Nutrients Terminology**
**Reviewer Comment:** "Line 429: 'The distribution of nutrients, i.e., dissolved organic matter (DOM),' DOM is not nutrients, but an indicator. Please correct."

**Response:** You are absolutely correct that dissolved organic matter (DOM) is not itself a nutrient; referring to it as such was a misstatement on our part. We have corrected the wording in the Discussion to eliminate this confusion. See revised sentence **on lines 484-489:**

> *"*The distribution of nutrients on the island has been shown to be linked to snowmelt and hill slope topography (Westergaard-Nielsen et al., 2020), which is likely playing a role during the early part of the season, but especially later in the year as DOM, *a proxy for nutrients*, becomes the primary limiting factor in predicting higher fluxes (Figure 7b-c) (Olid et al., 2021, 2022).*"*

**Line 494 — Clarification of pH/Oxygen Influence on CH₄ Emissions**
**Reviewer Comment:** "Line 494: 'that increasing pH and oxygen saturation as a result of primary production drive CH₄ emissions down through the growing season (Figure 7c-d and Figure A1).' This conclusion is not precise. Suggest to say 'increasing pH and oxygen saturation as a result of primary production and providing a vibrant aerobic environment, thus favored methanotrophic activities and then drive CH₄ emissions down through the growing season (Figure 7c-d and Figure A1).'?"

**Response:** We appreciate this suggestion and have adopted a revised wording very close to what the reviewer proposed to more precisely explain the mechanism by which higher pH and O₂ lead to lower CH₄ emissions. The sentence now reads **on lines 551-554**:

> *"*Here we show that increasing pH and oxygen saturation, as a result of primary production, create an aerobic environment that favors methanotrophic activity, thereby driving CH₄ emissions down through the growing season (Figure 7c-d, Figure A1).*"*

**Supplemental Material, Line 21 — "Least Mean Deviance Standard Error"**
**Reviewer Comment:** "In supplemental material, line 21: I feel really hard to interpret 'least mean deviance standard error'."

**Response:** We have revised the wording in the Supplementary Material to clarify this phrase. It was originally intended to describe the minimum mean deviance and its standard error from the BRT cross-validation (related to selecting the optimal number of trees). We realized the phrasing was confusing. See revised sentence **on line 29** of the Supplemental Material:

> *"*…the lowest mean deviance…*"*,

**Additional Clarifications:**

- Clarified that all BRT models were trained using 2024 data only on **lines 285-286**.

"Considering we only collected water temperature in 2023, we used flux data from 2024 for training the BRT."

Once again, we thank the reviewer for the thorough examination of both the main text and supplemental material. All the above changes have been incorporated in the revised manuscript and supplement, which we believe significantly improves clarity and accuracy. We trust that we have addressed all concerns, and we are grateful for the opportunity to improve our work based on Reviewer 2's insightful comments. Please let us know if any further clarifications are